
# SDUST2023VGGA: A Global Ocean Vertical Gradient of Gravity Anomaly Model Determined from Multidirectional Data from Mean Sea Surface

Ruichen Zhou[1,2,3], Jinyun Guo[1,*], Shaoshuai Ya[1], Heping Sun[2], and Xin Liu[1]

[1]College of Geodesy and Geomatics, Shandong University of Science and Technology, Qingdao 266590, China
[2]State Key Lab of Geodesy and Earth's Dynamics, Innovation Academy for Precision Measurement Science and Technology, Chinese Academy of Sciences, Wuhan 430077, China
[3]University of Chinese Academy of Sciences, Beijing 100049, China
[*]Correspondence: jinyunguo1@126.com

**Abstract.** Satellite altimetry is a vital tool for global ocean observation, providing critical insights into ocean gravity and its gradient. Over the past six years, satellite data from various space agencies have nearly tripled, facilitating the development of high-precision ocean gravity anomaly and ocean vertical gradient of gravity anomaly (VGGA) models. This study constructs a global ocean VGGA model named SDUST2023VGGA using multi-directional mean sea surface data. To address

5   computational resource limitations, the global ocean is divided into ten sub-regions. In each sub-region, the DTU21 Mean Sea Surface (MSS) model and the CNES-CLS22 Mean Dynamic Topography (MDT) model are used to derive the geoid. To mitigate the influence of long-wavelength signals on the calculations, the study subtracts the long-wavelength geoid derived from the XGM2019e gravity field model from the original geoid, resulting in a residual geoid (short-wavelength). To ensure the accuracy of the VGGA calculations, a weighted least-squares method is employed using residual geoid data from a 17′

10   × 17′ area surrounding the computation point. This approach effectively accounts for the actual ocean environment, thereby enhancing the precision of the calculation results. After combining the VGGA models for all sub-regions, the model's reliability is validated against the SIO V32.1 VGGA (named curv) model. The comparison between the VGGA and the SIO V32.1 model shows a mean is -0.08 Eötvös (E) and the RMS is 8.50 E, indicating a high degree of consistency across the global scale. Analysis of the differences reveals that the advanced data processing and modeling strategies employed in the DTU21 MSS

15   model enable SDUST2023VGGA to maintain stable performance across varying ocean depths, unaffected by ocean dynamics. The effective use of multi-directional mean sea surface data allows for the detailed capture of ocean gravity field information embedded in the MSS model. Analysis across diverse ocean regions demonstrates that the SDUST2023VGGA model successfully reveals the internal structure and mass distribution of the seafloor. The SDUST2023VGGA dataset is freely available at https://doi.org/10.5281/zenodo.14177000 (Zhou et al., 2024).

## 1   Introduction

Gradients of gravity are the partial derivatives of the gravity vector components along the three axes of a Cartesian coordinate system. They describe variations in the Earth's gravitational field in space, reflecting changes in both magnitude and direc-





tion. By enhancing high-frequency signals, gradients of gravity provide a more detailed representation of subsurface density structures (Mortimer, 1977). This capability makes them valuable for accurately depicting the spatial structure of field sources, understanding the Earth's internal structure, and identifying the location and depth of density variations (Romaides et al., 2001; Oruç, 2011; Panet et al., 2014). Consequently, gradients of gravity play a crucial role in geophysical exploration and ocean gravity field studies (Butler, 1984).

In recent years, ocean vertical gradient of gravity (VGG) have demonstrated significant potential across various disciplines, particularly in earthquake monitoring, underwater navigation, and ocean exploration. Fuchs et al. (2013) utilized gradient data from the GOCE satellite to detect substantial gravitational field changes resulting from the 2011 Tohoku earthquake in Japan, finding that the gravity signal exceeded model predictions. This study underscored the sensitivity and spatial correlation capabilities of gradient of gravity technology in capturing seismic signals. Similarly, Wang et al. (2023) proposed a wavelet-transform-based regional matching method that advanced the application of gradients of gravity in underwater navigation, significantly improving matching accuracy and demonstrating the technology's practicality and precision. Furthermore, gradient of gravity inversion has proven to be important in ocean exploration. Wan et al. (2023) combined satellite altimetry data with neural network techniques to predict global ocean depths, showcasing the broad applicability of gradients of gravity in oceanographic studies. Additionally, research by Kim and Wessel (2011) demonstrated that extracting short-wavelength information from satellite altimetry data effectively reveals the mass distribution of seafloor fracture zones and seamounts, offering new insights into geological structure analysis. These studies illustrate the diverse applications of gradients in earthquake monitoring, underwater navigation, bathymetry, and geological structure research.

As the application of gradient technology expands and gravity field theory evolves rapidly, gradient of gravity measurement techniques have seen continuous improvement (DiFrancesco et al., 2009; Stray et al., 2022; van der Meijde et al., 2015). However, measuring mobile gravity gradients remains costly, and achieving comprehensive global ocean coverage continues to be a challenge. The GOCE satellite, equipped with electrostatic accelerometer-based gravity gradiometers, provides global gravity data (Silvestrin et al., 2012; Rummel et al., 2011; Marks et al., 2013). Despite this, its resolution is limited in providing high-precision local gradient of gravity data (Novák et al., 2013).

In contrast, satellite altimetry technology has matured significantly and is now widely used to construct gravity potential models (Zingerle et al., 2020; Pavlis et al., 2012), mean sea surface (MSS) models (Andersen et al., 2023; Jin et al., 2011; Yuan et al., 2023), mean dynamic topography models (Knudsen et al., 2021; Jousset et al., 2023, 2022), ocean gravity models (Sandwell and Smith, 2009; Sandwell et al., 2014; Garcia et al., 2014; Hwang et al., 1998; Hao et al., 2023; Zhu et al., 2022; Li et al., 2024), ocean gradient of gravity models (Johannes Bouman and Sebera, 2011; Sandwell, 1992; Annan et al., 2024; Zhou et al., 2023), seafloor topography models (Smith and Sandwell, 1997; GEBCO Bathymetric Compilation Group 2024, 2024; Hu et al., 2020), sea level studies (Schwatke et al., 2015; Vignudelli et al., 2019; Ablain et al., 2017), and monitoring changes in terrestrial lake water levels (Hwang et al., 2016; Sulistioadi et al., 2015). Among these, the mean sea surface (MSS) is a relatively stable sea level, determined by averaging instantaneous sea surface height data from satellite altimetry over a specific time period (Andersen and Knudsen, 2009). MSS is crucial in both geodesy and physical oceanography, where it is extensively used to analyze ocean circulation, detect mesoscale eddies, assess sea level variations, determine geoid undula-





tions, and identify crustal deformation (Fu and Cazenave, 2001). MSS has broad applications across geodesy, oceanography, geophysics, and climatology. In geodesy, it serves as a global sea level reference, aiding in the study of geoid variations and assisting in determining the precise positions of Earth's surface features and vertical crustal movements. In oceanography, MSS is employed to study global ocean circulation, sea surface temperature, and changes in sea ice (Fu and Cheney, 1995; Vermeer and Rahmstorf, 2009; Skourup et al., 2017). In geophysics, it contributes to the analysis of the Earth's gravity field and seismic activity (Melini and Piersanti, 2006). In climatology, sea level data are fundamental for understanding the relationship between global sea level changes and climate change (Vermeer and Rahmstorf, 2009; Church and Gregory, 2001).

To fully extract the detailed ocean gravity field information embedded in the DTU21 MSS model, this study proposes a method for constructing a global vertical gradient of gravity anomaly (VGGA) model using multi-directional MSS data. Sect. 2 introduces the DTU21 MSS model and outlines the criteria for selecting additional datasets. Sect. 3 comprises two subsections: Sect. 3.1 outlines the strategy for partitioning the global ocean regions, describing how the ocean is divided into multiple areas based on geographic features and oceanographic dynamics. Sect. 3.2 presents a newly developed method aimed at maximizing the extraction of ocean gravity field information from the DTU21 MSS model, thereby enhancing the effectiveness of gravity field data extraction. Sect. 4 evaluates the constructed model and assesses its reliability, while Sect. 5 discusses the factors influencing the model's construction results and key findings observed during the process. After addressing data availability in Sect. 6, a summary of the model construction methods and results is provided in Sect. 7.

## 2 Research Data

### 2.1 DTU21MSS

The DTU21 MSS model was developed by the National Space Institute of the Technical University of Denmark (DTU) (Andersen et al., 2023). The dataset is publicly available at https://data.dtu.dk/articles/dataset/DTU21_Mean_Sea_Surface/19383221. This model utilizes satellite altimetry data from multiple missions, including Topex/Poseidon, the Jason series, CryoSat-2, and SARAL/AltiKa, providing high-precision sea surface observations from January 1, 1993, to December 31, 2012.

To enhance the precision and resolution of the data, the DTU21 MSS model employs a new processing chain that incorporates updated filtering and editing techniques. Compared to its predecessors, DTU15MSS and DTU18MSS, which were constructed using 1 Hz satellite altimetry data, DTU21 MSS is based on 2 Hz data, significantly improving its accuracy. The integration of satellite altimetry data with advanced retracking techniques and the application of the Parks-McClellan filtering algorithm in developing the DTU21 MSS model enabled enhanced resolution in the 10–40 km wavelength range, significantly improving accuracy in polar and coastal regions.

The DTU21 mean sea surface dataset is provided in a gridded format. By combining data from multiple satellite sources and utilizing advanced processing methods, the DTU21 model delivers high spatial resolution and precision on a global scale, making it a reliable resource for oceanographic and Earth science research.

## 2.2 CNES-CLS22 MDT

90 To extract geoid information from the mean sea surface (MSS) model, a Mean Dynamic Topography (MDT) model is required. The CNES-CLS22 MDT was derived by integrating data from satellite altimetry, the GRACE and GOCE gravity missions, and in-situ oceanographic measurements, such as drifter velocities, high-frequency radar velocities, and salinity-temperature profiles (Jousset et al., 2023). The dataset is accessible at https://www.aviso.altimetry.fr/en/data/products/auxiliary-products/mdt/mdt-global-cnes-cls.html.

95 Compared to its predecessor, CNES-CLS18 MDT, the CNES-CLS22 MDT demonstrates significant improvements in high-latitude regions. It offers broader coverage and eliminates artifacts, primarily due to the use of a new initial estimate that incorporates the CNES-CLS22 MSS and the GOCO06s geoid model. Additionally, optimal filtering techniques, such as Lagrangian filtering in coastal areas, were applied to further enhance accuracy.

Selecting the CNES-CLS22 MDT ensures minimal correlation with the DTU21 MSS (Knudsen et al., 2022, 2021), making 100 it more effective for extracting detailed ocean gravity field information embedded in the mean sea surface model. This choice enhances the credibility of the VGGA model developed in this study.

## 2.3 XGM2019e

A highly accurate Earth gravity field model is essential for applying the remove-restore method. In this study, the reference gravity field model XGM2019e was selected due to its widespread use in geoscience research and practical applications. The 105 model is available from the International Centre for Global Earth Models (ICGEM) at https://icgem.gfz-potsdam.de/home (Ince et al., 2019).

XGM2019e is a comprehensive global gravity field model, complete to degree and order 5399, offering a half-wavelength resolution of approximately 4 kilometers. It is primarily based on the GOCO06s satellite gravity field model, supplemented by a 15-minute terrestrial gravity dataset and a 1-minute enhanced gravity dataset provided by the U.S. National Geospatial-110 Intelligence Agency (Zingerle et al., 2020). The integration of satellite and ground-based gravity data ensures high precision in both large-scale and local gravity field representations, making XGM2019e suitable for detailed geophysical studies, including oceanography, tectonics, and geoid determination.

The model's high resolution and accuracy are critical for improving the reliability of the VGGA model developed in this study. By using XGM2019e in the remove-restore process, the long-wavelength components of the gravity field are efficiently 115 removed, allowing for focused extraction of the high-frequency VGGA from the mean sea surface model.

## 2.4 SIO V32.1

Given the significant challenges in obtaining in-situ measurements of VGGAs over the oceans, this study utilized the SIO V32.1 dataset, derived from satellite altimetry data, to validate its results. The SIO V32.1 dataset is a high-precision, high-resolution global ocean dataset that includes models for the VGGA, gravity anomaly, and the north-south and east-west components of 120 the deflection of the vertical (DOV) (Garcia et al., 2014). The latest version, V32.1, also incorporates an MSS model.



The SIO V32.1 dataset has been widely used in geophysical and oceanographic studies due to its ability to provide detailed gravity field information over oceanic regions, which is difficult to achieve using traditional measurement techniques. The combination of satellite altimetry data and advanced processing algorithms allows for high spatial resolution, making this dataset particularly useful for detecting fine-scale oceanic structures, including seamounts, fracture zones, and variations in seafloor topography. Using the SIO V32.1 dataset for cross-validation ensures the robustness and reliability of the VGGA model developed in this study.

## 2.5 GEBCO Bathymetric Model

To evaluate the performance of the SDUST2023VGGA model in different bathymetric conditions, this study employed the General Bathymetric Chart of the Oceans (GEBCO) bathymetric model as reference data. The GEBCO model is widely recognized in the field of geosciences for its detailed representation of the seafloor topography of the world's oceans. The model is available on the GEBCO project website ( https://www.gebco.net/data_and_products/gridded_bathymetry_data/).

The GEBCO bathymetric model combines ship-based bathymetric surveys and satellite-derived altimetric data. It provides a global, high-resolution depiction of seafloor depth. The current version, GEBCO2024, has a spatial resolution of 15 arc-seconds, which is approximately equivalent to 450 meters at the equator. This high resolution allows for the identification of various seafloor features such as seamounts, trenches, and ridges, thereby supporting a wide range of geophysical and oceanographic studies.

## 3 Research methods

### 3.1 Global zoning strategy

To address the limitations of computational resources, this study employed a partition-build-merge strategy. The global ocean was divided into several sub-regions, each computed independently before merging the results into a comprehensive model. For regions between latitudes 60° S and 60° N, each sub-region was defined as 30° × 30°. For regions outside these latitudes, sub-regions were set to 30° × 20° to account for the unique geographic characteristics of high-latitude areas. For easier analysis, the sub-regions were systematically labeled, as shown in Figure 1: horizontally, starting from 0° longitude and moving eastward, sub-regions were labeled L1 to L12; vertically, starting at 80° S latitude and moving northward, they were labeled B1 to B6.

To mitigate edge effects during calculations, a 2° boundary extension was applied to each sub-region. For sub-regions with a standard size of 30° × 30°, the MSS and MDT models were expanded to cover an area of 32° × 32°. After constructing the VGGA model for the extended 32° × 32° area, the 2° boundary was removed, leaving the core 30° × 30° sub-region, as shown in Figure 2. This clipping strategy ensured that boundary information was preserved during derivative calculations, effectively reducing edge effects and improving the accuracy of the final model. Extending sub-region boundaries before trimming minimized the loss of critical data at the edges, thereby ensuring more reliable results when computing the VGGA.

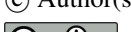


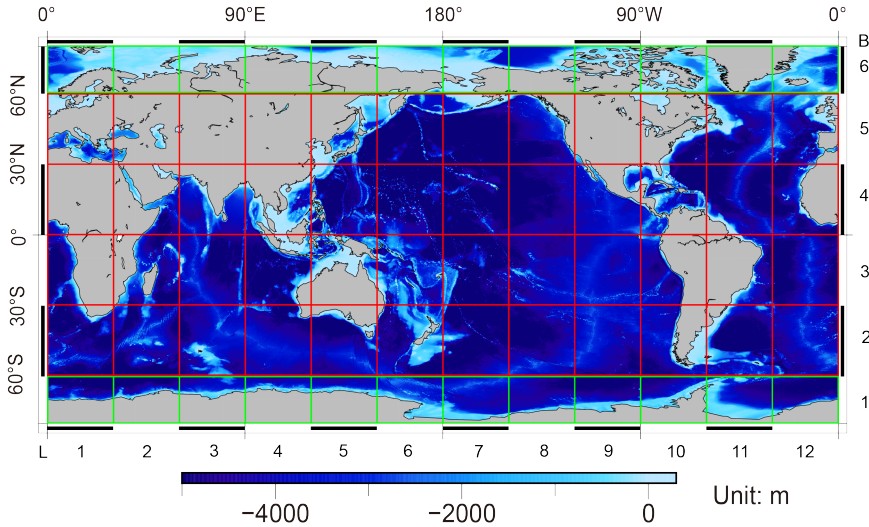

**Figure 1.** Global oceanic partition strategy using 30° × 30° and 30° × 20° grids based on the Partition-Build-Merge approach. The sub-regions are labeled as L1 to L12 (horizontal) and B1 to B6 (vertical) for identification.

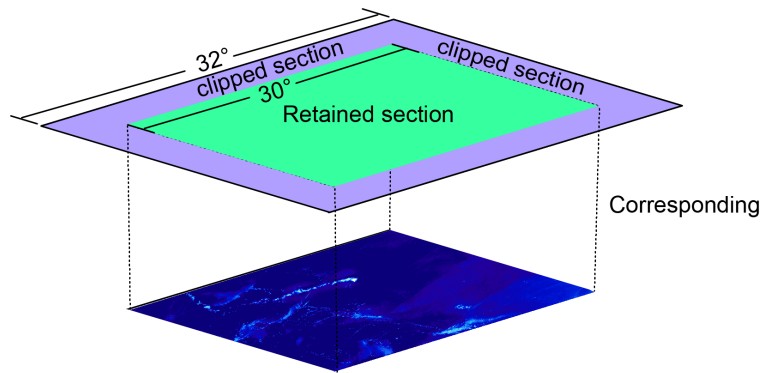

**Figure 2.** Clipping strategy with a 2° boundary extension to mitigate edge effects. Each 30° × 30° sub-region was expanded to 32° × 32° during computation, with the excess boundary removed after processing to ensure accuracy in the core region.

Addressing edge effects significantly enhances the precision of the VGGA model by ensuring that derivative calculations at boundaries are reliable.

## 3.2 Model building

The vertical gradient of gravity (VGG) is composed of two components: the vertical gradient of normal gravity and the vertical gradient of gravity anomaly (VGGA). The VGGA was initially derived using the MSS model. Since the vertical gradient of




normal gravity is a function of latitude, the VGGA alone is sufficient to reveal the Earth's internal structure. Therefore, only the VGGA was calculated in this study, without incorporating the vertical gradient of normal gravity.

The VGGA represents the difference between the gradient of actual gravity and the normal gravity, highlighting deviations caused by factors such as uneven mass distribution within the Earth, topographical variations, and subsurface structures. These

deviations provide valuable insights into subsurface geological formations and geophysical characteristics. By combining gravity anomalies with VGGA, a more detailed and comprehensive representation of the ocean gravity field can be achieved.

Assuming the disturbing potential, denoted as $T$, at a point on the geoid, the gravity anomaly is computed using the fundamental equations of physical geodesy (Moritz, 1980; Hofmann-Wellenhof and Moritz, 2006):

$$\Delta g = -\frac{\partial T}{\partial h} + \frac{1}{\gamma}\frac{\partial \gamma}{\partial h}T \tag{1}$$

where, $\Delta g$ represents the gravity anomaly, and $\gamma$ is the normal gravity at the point. If the ellipsoid is approximated as a sphere, the following approximation holds:

$$\frac{\partial}{\partial h} \cong \frac{\partial}{\partial r} \tag{2}$$

According to Bruns' formula, $\gamma$ can be replaced by $G$, resulting in the following relationship outside the Earth:

$$\Delta g = -\frac{\partial T}{\partial r} - \frac{2}{r}T \tag{3}$$

By differentiating the above equation and using Laplace's equation, a formula for calculating the VGGA, involving the geoid as well as its first and second horizontal derivatives, can be derived:

$$\frac{\partial \Delta g}{\partial r} = \frac{2G}{R^2}N - \frac{G}{R^2}\tan\varphi\frac{\partial N}{\partial \varphi} + \frac{G}{R^2}\frac{\partial^2 N}{\partial \varphi^2} + \frac{G}{R^2\cos^2\varphi}\frac{\partial^2 N}{\partial \lambda^2} \tag{4}$$

where $N$ represents the geoid, $G$ represents the mean gravity value, and $\varphi$ represents the geographic latitude of the calculation point.

After establishing the partitioning strategy and deriving the method for calculating the VGGA, the mean dynamic topography (MDT) model from CNES-CLS22 was interpolated onto the DTU21 grid using cubic spline interpolation to increase the resolution. This increased the resolution to $1' \times 1'$ and ensured consistency between the two datasets. Subsequently, the interpolated MDT model was subtracted from the DTU21 MSS model to obtain the geoid, denoted by $N$:

$$N = MSS - MDT_{\text{inter}} \tag{5}$$

Next, the remove-restore method was used to isolate the residual geoid signal by subtracting the long-wavelength component, represented by the XGM2019e geoid, from the original geoid. This step separates the long-wavelength and short-wavelength components of the geoid. By removing the long-wavelength signals, the short-wavelength residual geoid remains, highlighting local variations and finer details in the geoid structure (Hwang, 1999).

$$N_{\text{res}} = N - N_{\text{ref}} \tag{6}$$



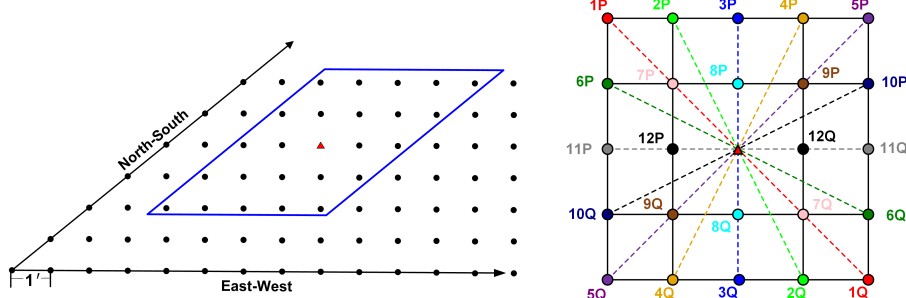

**Figure 3.** Illustration of the north-south and east-west components of the derivative of the residual geoid based on multidirectional MSS data, using a $5' \times 5'$ window as an example. The red triangle represents the center point for calculation, and P-Q pairs indicate 12 possible combinations used for multi-directional gradient computation.

Through the remove-restore method, the gridded MSS provided a gridded form of the residual geoid, $N_{\text{res}}$. For each calculation point, the second-order derivatives of $N_{\text{res}}$ were computed in multiple directions using a least squares method. The second-order partial derivatives in the north-south and east-west components were then computed to optimize the extraction of gravity field information, ensuring accuracy and reliability. This approach takes into account the complexity of ocean topography to enhance gravity field information extraction, ensuring both accuracy and reliability. The following formula is used for

calculation:

$$\frac{\partial^2 N_{\text{res},(kP,kQ)}}{\partial l^2} = \frac{\partial^2 N_{\text{res},(kP,kQ)}}{\partial \varphi^2} \cos^2 a_{kP,kQ} + 2\frac{\partial^2 N_{\text{res},(kP,kQ)}}{\partial \varphi \partial \lambda} \sin a_{kP,kQ} \cos a_{kP,kQ} + \frac{\partial^2 N_{\text{res},(kP,kQ)}}{\partial \lambda^2} \sin^2 a_{kP,kQ} \tag{7}$$

where $k$ represents the number of equations, and $\alpha$ represents the geodetic azimuth between points $P$ and $Q$. According to the method in (Zhou et al., 2023), a calculation window size of $17' \times 17'$ was determined. For a given calculation window size of $i' \times i'$, the number of equations is calculated as: $k = (i^2 - 1)/2$. For example, when i = 5, 12 sets of second-order derivatives

of $N_{\text{res}}$ are obtained, resulting in 12 equations. Figure 3 illustrates the calculation process using a $5' \times 5'$ calculation window. In the actual computations, a $17' \times 17'$ window was used.

The matrix form of Equation 7 can be represented as:

$$\mathbf{V} = \mathbf{AX} - \mathbf{L} \tag{8}$$

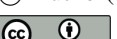

where $\mathbf{V}$ represents the residual vector, and $\mathbf{A}$ represents the coefficient matrix composed of north-south and east-west second-order partial derivatives:

$$\mathbf{A} = \begin{pmatrix} \cos^2 \alpha_{1P,1Q} & 2\sin \alpha_{1P,1Q} \cos \alpha_{1P,1Q} & \sin^2 \alpha_{1P,1Q} \\ \cos^2 \alpha_{2P,2Q} & 2\sin \alpha_{2P,2Q} \cos \alpha_{2P,2Q} & \sin^2 \alpha_{2P,2Q} \\ \cos^2 \alpha_{3P,3Q} & 2\sin \alpha_{3P,3Q} \cos \alpha_{3P,3Q} & \sin^2 \alpha_{3P,3Q} \\ \vdots & \vdots & \vdots \\ \cos^2 \alpha_{(k-1)P,(k-1)Q} & 2\sin \alpha_{(k-1)P,(k-1)Q} \cos \alpha_{(k-1)P,(k-1)Q} & \sin^2 \alpha_{(k-1)P,(k-1)Q} \\ \cos^2 \alpha_{kP,kQ} & 2\sin \alpha_{kP,kQ} \cos \alpha_{kP,kQ} & \sin^2 \alpha_{kP,kQ} \end{pmatrix} \tag{9}$$

$\mathbf{X}$ consists of two vectors representing the second-order partial derivatives of the residual geoid in the north-south and east-west directions. Once the residual vector and coefficient matrix are established, the next step involves applying the weighting matrix. The solution for $\mathbf{X}$, corresponding to the north-south and east-west components of the second-order partial derivatives of the geoid, is determined by minimizing the cost function: $\Psi = V^T W V$:

$$\mathbf{X} = (\mathbf{A}^T \mathbf{W} \mathbf{A})^{-1}(\mathbf{A} \mathbf{W} \mathbf{V}) \tag{10}$$

where $W$ is the weighting matrix, calculated using an inverse distance formula: $W = 1/l^2$. This formula gives higher weights to closer points, thereby balancing data contributions and ensuring accuracy and reliability in the results.

Unlike this study, which calculates the VGGA by deriving the second-order horizontal partial derivatives of the geoid, the VGGA model in the SIO V32.1 dataset is computed from the first-order derivatives of the DOV's north-south and east-west components (Muhammad et al., 2010; Sandwell and Smith, 1997, 2009). By contrast, the VGGA model relies on first-order derivative calculations. To evaluate and validate the accuracy of the proposed method, the model constructed in this study was compared with the SIO V32.1 dataset.

The workflow of the study is illustrated in Figure 4.

## 4 Results and analysis

After establishing the partitioning strategy, the DTU21 MSS was used, and the CNES-CLS22 MDT was subtracted to derive the geoid. Subsequently, the XGM2019e gravity field was subtracted to obtain the residual geoid. A multi-directional method was applied to the MSS data to calculate the second-order derivatives, and the VGGA) for each sub-region was estimated through the least squares method. The sub-regional VGGA models were then combined to construct a global VGGA model for oceanic regions.

Due to the limited availability of direct global VGGA observations, the satellite altimetry-based SIO V32.1 curv model served as a reference for comparison. The inversion statistics for each sub-region are presented in Table 1 presents the inversion statistics for each sub-region, highlighting discrepancies between the VGGA models. The differences between the two models were analyzed to assess and validate the accuracy of the computational method used in this study.



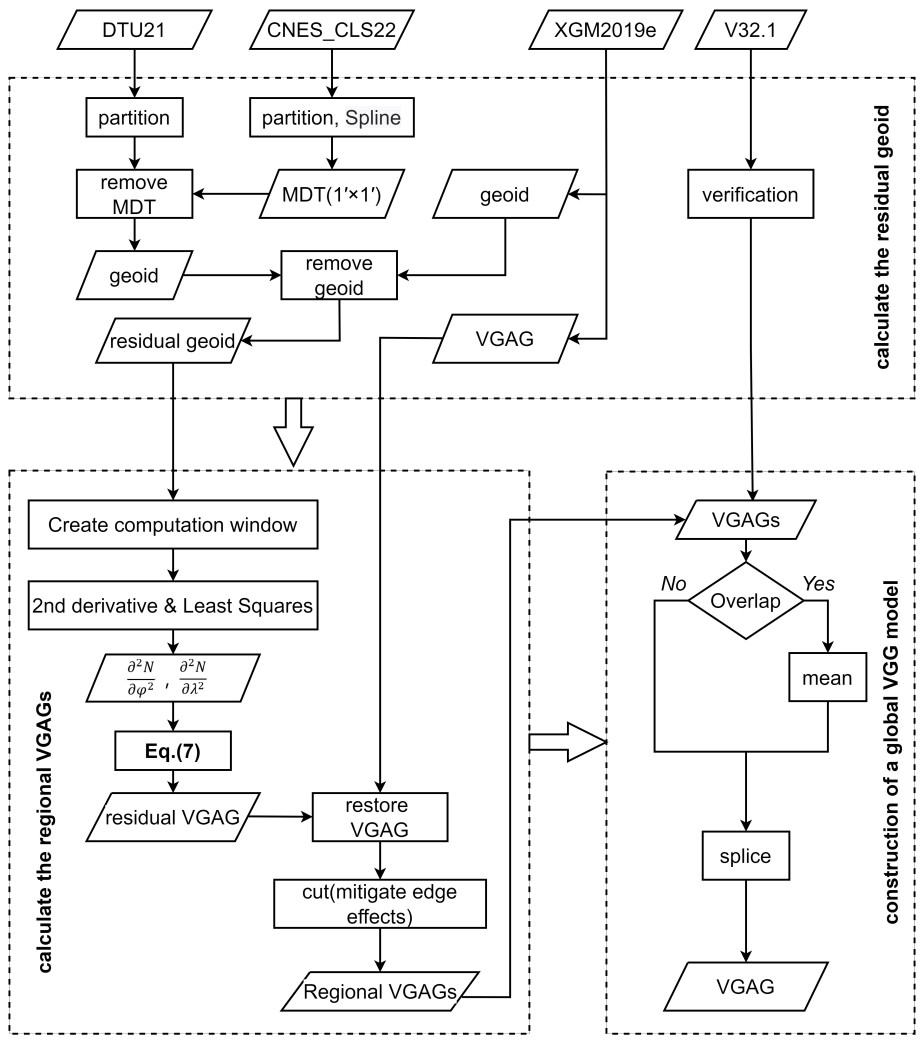

**Figure 4.** Workflow for constructing the VGGA model using the DTU21 MSS, CNES-CLS22 MDT and XGM2019e.

Table 1 presents a statistical assessment of the root mean square (RMS) differences between the constructed VGGA models and the SIO V32.1 curv model for each sub-region.

The analysis of the longitudinal direction (L1 to L12) shows significant fluctuations without a clear trend, indicating that longitudinal VGGAs are affected by various geographical and geophysical factors, such as variations in seafloor topography and local tectonics. In the latitudinal direction (B1 to B6), particularly in the equatorial regions (B3 and B4), lower anomaly

**Table 1.** Statistical RMS differences obtained by subtracting the constructed VGGA models from the SIO V32.1 curv model (unit: E)

|      | B1    | B2   | B3    | B4    | B5    | B6    |
|------|-------|------|-------|-------|-------|-------|
| L1   | 10.03 | 4.55 | 5.67  | 5.60  | 6.12  | 9.91  |
| L2   | 11.44 | 5.06 | 6.03  | 4.24  | 7.88  | 11.82 |
| L3   | 10.24 | 5.43 | 5.65  | 6.82  | 17.99 | 18.76 |
| L4   | 14.92 | 8.97 | 8.58  | 11.69 | 9.13  | 16.92 |
| L5   | 20.64 | 5.45 | 6.24  | 5.43  | 6.14  | 14.39 |
| L6   | 12.88 | 5.96 | 4.76  | 6.54  | 7.10  | 12.63 |
| L7   | 11.98 | 5.76 | 7.02  | 9.38  | 12.59 | 10.41 |
| L8   | 13.10 | 5.94 | 8.49  | 10.35 | 68.39 | 6.83  |
| L9   | 11.55 | 5.78 | 6.02  | 6.90  | 64.41 | 7.72  |
| L10  | 16.63 | 5.26 | 7.12  | 10.35 | 38.25 | 11.08 |
| L11  | 14.43 | 5.62 | 14.62 | 9.63  | 9.98  | 9.33  |
| L12  | 17.00 | 6.66 | 8.74  | 6.04  | 6.35  | 8.71  |

values were observed, likely due to the high quality of altimetry data in these areas. Conversely, higher RMS values were noted in sub-region B6, especially in L3, L4, and L5, indicating the influence of more complex bathymetric features and tectonic activity typical of sub-polar regions.

    Further analysis, along with the ocean depth model in Figure 1, suggests that significant VGGAs near the equator are linked to deep ocean trenches, mid-ocean ridges, and other geological structures that notably affect the Earth's mass distribution.
Additionally, longitudinal sub-regions such as L1, L2, and L6 exhibit lower RMS differences across most latitude bands, indicating potentially more homogeneous geoid characteristics in these longitudinal strips.

    Upon completing the model construction for all sub-regions, excess portions were trimmed to create the global VGGA model for oceanic regions, named SDUST2023VGGA. The model spans longitudes from 0° to 360° and latitudes from 80° S to 80° N, as illustrated in Figure 5.

Investigating Earth's tectonic activities and their impact on gravitational fields enhances our understanding of geophysical dynamics. Supported by the visualizations in Figure 5, this study examines key tectonic features, including mid-ocean ridges, abyssal plains, subduction zones, and volcanically active zones. These elements are pivotal in shaping Earth's landscape and influencing the observed patterns of VGGAs. The discussions that follow will analyze how these features contribute to gravity field variations and their implications for understanding underlying geodynamic processes.

Mid-ocean ridges serve as prominent features representing divergent boundaries in the global plate tectonic system. Notable examples include the Mid-Atlantic Ridge and the East Pacific Rise, both of which are characterized by mantle upwelling, where magma rises and solidifies to form new oceanic crust (Williams et al., 2008). This process results in localized density reduction. This creates a stark contrast with the surrounding older, cooler oceanic crust, leading to pronounced variations in



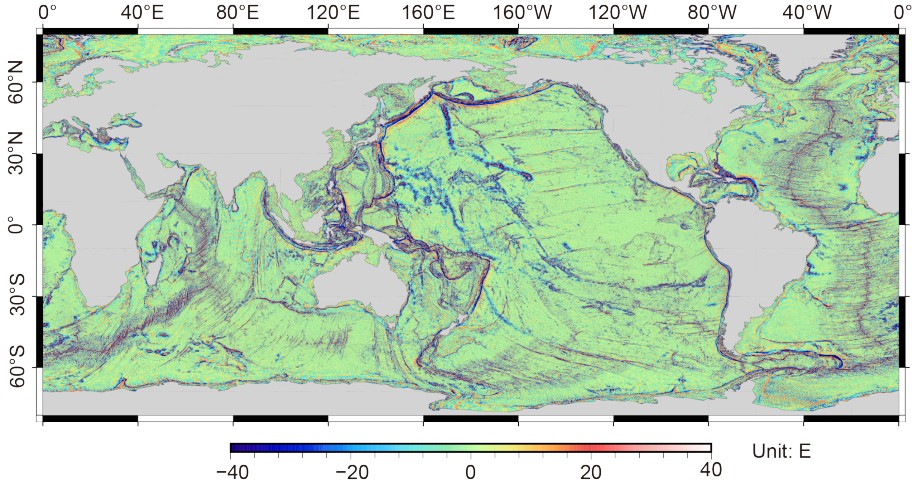

**Figure 5.** Global VGGA model SDUST2023VGGA constructed from the DTU21 MSS data.

the gravity field (Álvarez et al., 2018). Mantle upwelling influences the VGGA along mid-ocean ridges by reducing density,
reflecting crustal expansion and the formation of new oceanic material.

For instance, the Mid-Atlantic Ridge extends southward from Iceland 65° N, 18° W, passes through the North Atlantic 45°
N, 30° W, and reaches the South Atlantic 40° S, 10° W (Rao et al., 2004). On the gradient map, this ridge appears as a linear
high-gradient region, reflecting crustal changes due to mantle upwelling and expansion. Similarly, the East Pacific Rise, a
volcanically active chain located in the eastern Pacific, now extends from 55° S, 130° W and trends southwestward. This ridge
is characterized by significant volcanic activity, hydrothermal vents, and mantle upwelling, contributing to crustal expansion
in this area (Yu et al., 2022). The Central Pacific Region, extending from 10° N to 30° N and 140° W to 170° W, also features
significant volcanic activity and crustal thickness variations due to ongoing volcanic and tectonic processes (Rao et al., 2004).
This region's gravity anomalies are influenced by the uplift of the Hawaiian volcanic chain, exemplifying the complex interplay
of tectonic and volcanic dynamics (Poland and Carbone, 2016).

Deep-sea plains, typically linked to plate subsidence, where the oceanic crust gradually sinks under its weight, result in
relatively stable gradient (Hieke et al., 2003). The Pacific Abyssal Plain, extending from approximately 30° N to 60° S and
spanning longitudes from 150° W to 150° E, is characterized by a relatively thin crust and smooth, low-gradient regions on the
map. This uniformity is influenced by low-density, fine-grained sediments, such as silt and clay, that cover the plain, minimizing
variations in crustal density (Zhang et al., 2016). Similarly, the Atlantic Abyssal Plains, primarily distributed between 40° N
and 60° S and from 30° W to 30° E, display low gradient of gravity anomaly due to similar sedimentary characteristics,
which dampen tectonic activity and stabilize the crust. The Indian Ocean Abyssal Plains, located between 30° S and 60°
S and spanning longitudes from 60° E to 100° E, exhibit comparable low-gradients, highlighting the effects of thick, fine-
grained sediment deposits in maintaining crustal stability. These examples illustrate the global distribution of abyssal plains as
tectonically stable regions, with low-density sediments contributing to their distinct gradient of gravity anomaly patterns.



Subduction zones are regions where tectonic processes cause dramatic changes in crustal density, leading to distinct low VGGAs. For instance, the Mariana Trench, located at 11° N, 142° E, is the world's deepest oceanic trench, with the Challenger Deep reaching a depth of 10,994 meters. At these depths, the pressure increases water density by nearly 5%, contributing to significant gravity field variations, represented as dark blue areas indicating low VGGAs (Kim et al., 2009). Similarly, the Peru-Chile Trench, situated at approximately 23° S, 71° W, marks the subduction of the Nazca Plate beneath the South American

Plate, which generates significant seismic and volcanic activity, contributing to notable low VGGAs (Álvarez et al., 2018; Yang and Fu, 2018).

    Volcanically active zones exhibit notable VGGAs caused by crustal uplift, fractures, and magma activity. These regions often show localized high gradient of gravity anomaly due to magma chambers and fault zones. For instance, the Hawaiian Volcanic Chain, which spans from approximately 18° N to29° N and 155° W to 172° W, is a hotspot-driven volcanic chain

with notable gravity anomalies caused by magma upwelling. Similarly, the Galápagos Volcanic Chain, located from 0° N, 89° W to 1° N, 92° W, experiences significant volcanic activity due to the Galápagos hotspot, with density variations contributing to localized high gravity anomalies (Vigouroux et al., 2008). The New Zealand Volcanic Arc, located at the convergence of the Pacific and Indo-Australian Plates (35° S to 39° S, 175° E to 179° E), shows similar high-gradient associated with subsurface magma activity and crustal thickness variations. These volcanic zones, especially in regions like the Central Pacific Region,

illustrate the complex interactions between tectonic activity and mantle processes.

    The VGGA map reveals crucial density variations within the crust and mantle, providing an indispensable tool for identifying geological structures, exploring mineral resources, and assessing seismic and volcanic risks. The findings from this analysis emphasize the utility of VGGAs in elucidating tectonic and volcanic processes on a global scale.

    Given the challenges in obtaining direct VGGA measurements, the reliability of the VGGA model was assessed by com-

parison with the SIO V32.1 curv model. The difference between the VGGA model and the SIO V32.1 curv model produced a residual model, illustrated in Figure 6. The SIO V32.1 curv model, based on satellite altimetry data, serves as an effective reference for cross-validation.

    Figure 6 presents the global spatial distribution of differences between the VGGA model and the SIO V32.1 curv model. In most regions, the two models demonstrate strong consistency, exhibiting generally low residuals across the oceans. This

consistency is particularly evident in mid and low latitudes, where satellite altimetry data tends to be more accurate and less affected by atmospheric or sea-ice interference.

    Furthermore, increased differences between the models are evident from 66° S to 80° S, with a distinct seam appearing at 66° S. This seam likely arises from differences in data quality and processing methods applied to polar regions, underscoring the challenges in these areas. These challenges arise from factors such as limited satellite coverage, interference from sea ice,

and the need for distinct data calibration approaches in high-latitude zones. Such systematic variations in preprocessing may contribute to the observed discrepancies.

    To mitigate the impact of outliers and better assess data quality, a two-step filtering process was applied in the analysis of differences between the two models. Here, $\sigma$ represents the standard deviation (STD), measuring the dispersion of data points from the mean. First, a tenfold STD ($10\sigma$) filter was applied, which removed 0.07% of the data, reducing extreme outliers. Then,


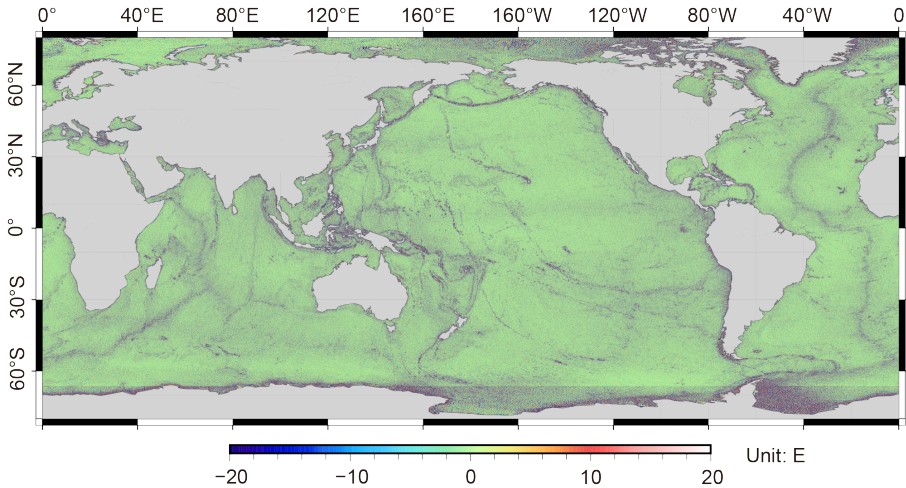

**Figure 6.** The spatial distribution of differences obtained by subtracting the constructed VGGA model from the SIO V32.1 curv model

**Table 2.** Statistical differences obtained by subtracting the constructed VGGA model from the SIO V32.1 curv model (unit: E)

| Data | Max | Min | Mean | STD | RMS | Exclusion Rate |
|------|-----|-----|------|-----|-----|----------------|
| difference | 791.01 | -673.00 | -0.08 | 7.24 | 8.50 | - |
| $10\sigma$ filter | 72.31 | -72.47 | -0.09 | 6.39 | 7.61 | 0.07% |
| $3\sigma$ filter | 21.64 | -21.79 | -0.07 | 4.82 | 5.85 | 1.41% |

a stricter threefold STD ($3\sigma$) filter was applied, which removed an additional 1.41% of the data, targeting moderate outliers. This hierarchical filtering approach refined the residuals by excluding outliers, facilitating a more detailed and accurate analysis of the model's accuracy and robustness. The statistical information and histogram of the residual differences are presented in Table 2 and Figure 7, respectively.

     Globally, the VGGA model derived from multi-directional MSS data shows strong consistency with the SIO V32.1 curv

model. Initial statistical analysis of the model differences revealed a maximum of 791.01 E and a minimum of -673.00 E, with a mean of -0.08 E, indicating an approximately symmetric distribution centered around zero. The STD was 7.24 E, and the root mean square (RMS) error was 8.50 E, suggesting the presence of outliers, potentially resulting from satellite altimetry limitations in coastal areas and regions with complex seafloor topography.

     To assess the impact of outliers on the statistical metrics, data points deviating by more than ±10 and ±3 times the STD

from the mean were progressively removed, followed by a re-analysis of the remaining dataset. After excluding data points exceeding ±10 times the STD, the maximum and minimum values were reduced to 72.31 E and -72.47 E, respectively, while the mean remained at -0.09 E. This indicates that extreme outliers significantly impacted the STD and RMS error; however, since these outliers represented only 0.07% of the total data, their effect on the mean was minimal.

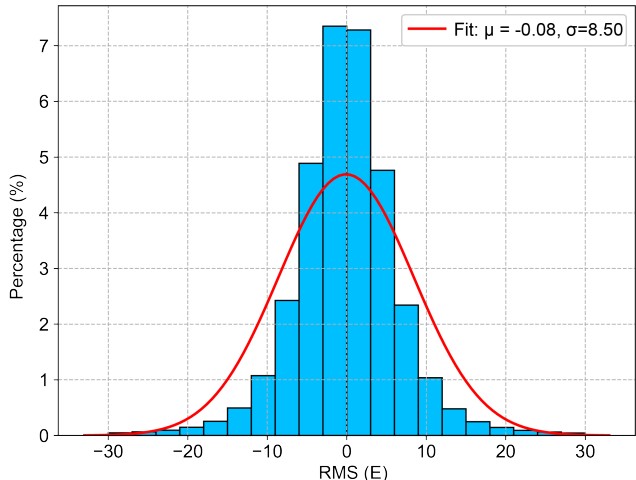

**Figure 7.** The histogram of differences obtained by subtracting the constructed VGGA model from the SIO V32.1 curv model, with fit parameters: $\mu$ representing the mean and $\sigma$ representing the STD

Further exclusion of data points deviating by more than $\pm 3$ times the STD (1.41% of the data) reduced the maximum and minimum values to 21.64 E and -21.79 E, with the mean shifting slightly to -0.07 E. This demonstrates that removing more outliers led to a continued decrease in the STD and RMS error, indicating reduced data dispersion and an improved fit. Nevertheless, the stability of the mean suggests that the core results were robust despite the presence of outliers. This stability implies that the underlying model retains its predictive capacity and reliability, even when outliers are present, which is crucial for ensuring the robustness of the VGGA model in diverse oceanic regions.

## 5 Discussion

Coastlines and islands can contaminate altimetry waveform data (Guo et al., 2010), which significantly affects the accuracy of altimetry measurements and, consequently, the inversion results of gradients of grvity anomaly. In deep-sea regions, complex ocean dynamics such as deep currents, internal waves, and eddies substantially impact sea surface morphology by causing undulations and localized changes in sea surface height (Khaki et al., 2015). Additionally, the satellite's orbital inclination affects its coverage of different latitudinal regions, potentially resulting in higher data quality at mid-latitudes and reduced coverage at the poles (Sandwell et al., 2006), thus influencing the quality of altimetry data. Furthermore, the slope of the seafloor topography plays a crucial role in the accuracy of altimetry data (Sandwell and Smith, 2014). Steeper slopes increase the complexity of sea surface morphology, resulting in more irregular altimetry waveforms and reduced data precision. Larger seafloor slopes increase the complexity of sea surface morphology, resulting in irregular signals in the altimetry waveforms. This reduces the precision of the data, thus affecting the reliability of the models constructed from these measurements. Coastlines and islands can contaminate altimetry waveform data (Guo et al., 2010), which significantly affects the accuracy of altimetry measure-

**Table 3.** Statistical summary of differences obtained by subtracting the VGGA model from SIO V32.1 curv at different offshore distances (unit: E)

| Offshore Distance (km) | Max | Min | Mean | STD | RMS |
|---|---|---|---|---|---|
| $[0, 50)$ | 791.01 | -673.00 | -0.81 | 18.37 | 18.39 |
| $[50, 100)$ | 247.45 | -163.67 | -0.08 | 7.55 | 7.55 |
| $[100, 150)$ | 187.57 | -119.80 | -0.05 | 6.54 | 6.54 |
| $[150, 200)$ | 151.67 | -102.02 | -0.05 | 6.21 | 6.21 |
| $[200, 250)$ | 133.24 | -88.72 | -0.04 | 6.04 | 6.04 |
| $[250, 300)$ | 130.08 | -129.01 | -0.03 | 5.88 | 5.88 |
| $[300, \infty)$ | 249.47 | -149.96 | -0.03 | 5.29 | 5.29 |

**Table 4.** Statistical summary of differences obtained by subtracting the VGGA model from SIO V32.1 curv at different ocean depths (unit: E)

| Depth (km) | Max | Min | Mean | STD | RMS |
|---|---|---|---|---|---|
| $[0, 1)$ | 791.01 | -671.45 | -0.20 | 13.39 | 13.39 |
| $[1, 2)$ | 709.38 | -442.57 | -0.01 | 10.05 | 10.05 |
| $[2, 3)$ | 315.07 | -492.87 | 0.07 | 7.79 | 7.79 |
| $[3, 4)$ | 262.21 | -557.07 | -0.07 | 6.03 | 6.03 |
| $[4, 5)$ | 134.04 | -162.81 | -0.11 | 4.83 | 4.83 |
| $[5, \infty)$ | 42.69 | -93.17 | -0.16 | 4.21 | 4.22 |

ments and, consequently, the inversion results of gradients of gravity anomaly. In deep-sea regions, complex ocean dynamics such as deep currents, internal waves, and eddies substantially impact sea surface morphology by causing undulations and localized changes in sea surface height (Khaki et al., 2015). Additionally, the satellite's orbital inclination affects its coverage
of different latitudinal regions, potentially resulting in higher data quality at mid-latitudes and reduced coverage at the poles (Sandwell et al., 2006), thus influencing the quality of altimetry data. Furthermore, the slope of the seafloor topography plays a crucial role in the accuracy of altimetry data (Sandwell and Smith, 2014). Steeper slopes increase the complexity of sea surface morphology, resulting in more irregular altimetry waveforms and reduced data precision. Larger seafloor slopes increase the complexity of sea surface morphology, resulting in irregular signals in the altimetry waveforms. This reduces the precision of
the data, thus affecting the reliability of the models constructed from these measurements.

To evaluate the impact of these factors on the inversion results, this study uses the residual model obtained in Sect. 4, calculated by subtracting the VGGA model from the SIO V32.1 curv model. The residuals are categorized into several intervals based on offshore distance, sea depth, latitude, and seafloor topography slope to examine the influence of these factors on model construction results. The findings are presented in Tables 3 through 6 and Figure 8.




**Table 5.** Statistical summary of differences obtained by subtracting the VGGA model from SIO V32.1 curv at different latitudes (unit: E)

| Latitude (°) | Max | Min | Mean | STD | RMS |
|---|---|---|---|---|---|
| $[-60,-50)$ | 670.36 | -339.30 | -0.02 | 5.73 | 5.73 |
| $[-50,-40)$ | 791.01 | -673.00 | -0.04 | 5.91 | 5.91 |
| $[-40,-30)$ | 262.57 | -193.64 | -0.05 | 5.96 | 5.96 |
| $[-30,-20)$ | 267.98 | -257.79 | -0.05 | 6.25 | 6.25 |
| $[-20,-10)$ | 258.93 | -259.24 | -0.07 | 6.72 | 6.72 |
| $[-10,0)$ | 397.26 | -650.42 | -0.15 | 8.11 | 8.11 |
| $[0,10)$ | 463.47 | -415.12 | -0.09 | 7.44 | 7.44 |
| $[10,20)$ | 479.81 | -557.07 | -0.15 | 7.46 | 7.46 |
| $[20,30)$ | 351.52 | -554.68 | -0.11 | 7.60 | 7.60 |
| $[30,40)$ | 353.69 | -529.66 | -0.20 | 7.37 | 7.38 |
| $[40,50)$ | 425.17 | -255.52 | -0.21 | 7.31 | 7.31 |
| $[50,60)$ | 668.93 | -504.54 | -0.17 | 8.78 | 8.78 |
| $[60,70)$ | 677.85 | -409.98 | -0.19 | 12.43 | 12.44 |
| $[70,80]$ | 428.13 | -219.00 | -0.05 | 11.40 | 11.41 |

**Table 6.** Statistical summary of differences obtained by subtracting the VGGA model from SIO V32.1 curv at different seafloor slopes (unit: E)

| Slope (%) | Max | Min | Mean | STD | RMS |
|---|---|---|---|---|---|
| $[0,1)$ | 682.50 | -641.65 | -0.17 | 7.18 | 7.19 |
| $[1,2)$ | 668.85 | -671.45 | -0.10 | 6.47 | 6.47 |
| $[2,3)$ | 740.85 | -673.00 | -0.07 | 6.65 | 6.66 |
| $[3,4)$ | 754.55 | -650.42 | -0.07 | 7.06 | 7.06 |
| $[4,5)$ | 668.93 | -446.56 | -0.06 | 7.50 | 7.50 |
| $[5,6)$ | 764.00 | -464.99 | -0.04 | 7.93 | 7.93 |
| $[6,7)$ | 758.45 | -509.98 | -0.02 | 8.36 | 8.36 |
| $[7,8)$ | 638.45 | -412.81 | 0.01 | 8.73 | 8.73 |
| $[8,9)$ | 791.01 | -366.35 | 0.02 | 9.11 | 9.11 |
| $[9,10)$ | 768.50 | -521.05 | 0.03 | 9.60 | 9.60 |
| $[10,\infty)$ | 768.05 | -557.07 | 0.05 | 12.32 | 12.32 |

Table 3 illustrates the impact of offshore distance on the inversion results. As the distance from the coastline increases, the STD of the residual model consistently decreases. Notably, beyond 50 kilometers from the coastline, the reduction in the STD becomes less pronounced. This is likely due to the reduced impact of shallow waters on satellite echo waveforms, which


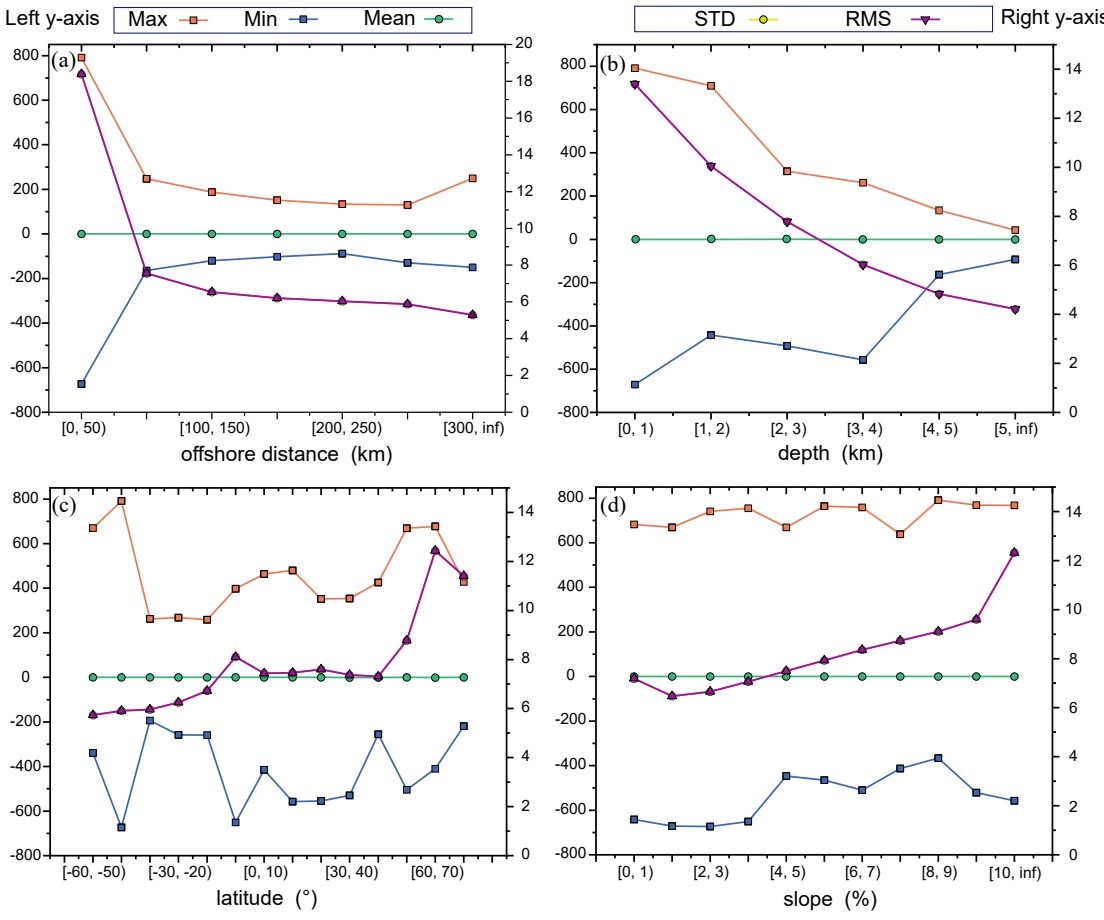

**Figure 8.** Residuals obtained by subtracting the constructed VGGA model from the SIO V32.1 curv model, categorized by: (a) offshore distance (km), (b) seafloor slope (%), (c) sea depth (m), and (d) latitude (°). The left y-axis represents the max, min, and mean residuals (unit: E), while the right y-axis represents the STD and RMS of the residuals (unit: E). Note: STD and RMS curves overlap, appearing as a single line.

results in less pronounced improvements in data accuracy. In summary, regions closer to the coastline are associated with larger residual values, while deep-sea regions farther from the coast exhibit lower residual values.

Table 4 provides statistical data of residuals across different sea depth intervals. In shallow waters (0-1 km), the residuals show considerable fluctuation, including extreme maximum and minimum values, as well as higher STD and RMS errors. These fluctuations may result from rapid seafloor topography changes in shallow regions, which substantially impact the vertical gradient of gravity anomaly calculations. As depth increases, the residual values gradually become smaller and more concentrated, indicating that the model performs more stably and reliably in deep-water areas. This stability can be attributed to

the relatively smooth seafloor topography in deeper waters, which introduces fewer interference factors. These results validate





that the DTU21 MSS model not only effectively handles oceanic phenomena, such as eddies, but also demonstrates a high level of completeness and robustness in capturing sea surface variability across different ocean environments.

The analysis of Table 5 shows that in the latitude intervals of [-60, -50)°, [-50, -40)°, [-10, 0)°, [60, 70)°, and [70, 80]°, the model exhibits larger extreme values, with maxima reaching 670.36 E, 791.01 E, 397.26 E, 677.85 E, and 428.13 E,
respectively. These high values indicate significant inconsistencies between models in these regions, especially in the high-latitude intervals such as [60, 70)° and [70, 80)°. The STD and RMS values are significantly higher in these intervals, at 12.43 E and 11.40 E, respectively, further confirming the greater inconsistencies in high-latitude regions. In contrast, mid- and low-latitude regions exhibit smaller extreme value ranges and lower RMS values, indicating greater model consistency. These results suggest that in mid- and low-latitude regions, the differences between the two models are smaller, indicating overall higher
reliability. Considering the orbital inclination of altimetry satellites and their measurement methods, these discrepancies can be attributed to the complexity of geological structures in high-latitude regions. Furthermore, areas near coastlines and shallow waters show greater model differences, increasing the discrepancies between models. The mean values across all latitude intervals approach zero, demonstrating strong global consistency between the two models. This supports the effectiveness of utilizing multi-directional mean sea level data for global gradient of gravity anomaly inversion.

Table 6 examines the impact of seafloor slope on the residual VGGA model. The results show that the STD and RMS values also show an increasing trend with greater slopes. For instance, in regions with slopes less than 1%, the RMS is relatively low at 7.19 E, indicating higher model stability and consistency. However, in areas where the slope exceeds 10%, the RMS increases to 12.32 E, suggesting a decrease in model reliability. This observation indicates that as the seafloor becomes steeper, it introduces more interference factors that challenge the reliability of VGGA inversion.

To further validate the study's conclusions, five representative regions were selected for model difference analysis. The topographic information of these regions is presented in Figure 9, providing a comprehensive understanding of the terrain characteristics that may influence model differences.

The selected regions include the open deep-sea area of the South Pacific far from land (Region A), the southeastern Atlantic region near the continental shelf with varied topography (Region B), the area near the Indonesian archipelago with complex
seafloor topography (Region C), the high-latitude North Atlantic region near the Arctic Circle (Region D), and the western Pacific deep-sea region characterized by complex underwater terrain but well-covered by satellites (Region E). A comparative analysis of these regions explored the effects of offshore distance, latitude, and sea depth on altimetry inversion accuracy of altimetry inversion, with the statistics provided in Table 7.

The modeling results in different regions are shown in Figure 10.

Region A in the South Pacific (5° S to 5° N, 120° W to 110° W) is located in a remote, open deep-sea area, far from any landmasses or islands. The seafloor topography in this region is predominantly flat, with no significant seamounts or trenches. These conditions create a stable environment for satellite altimetry data inversion, leading to minor residuals between the VGGA and SIO V32.1 models. The low STD and RMS indicate strong consistency between the models in this region. This suggests that, in open ocean areas distant from land, the model performs with high accuracy and stability, likely due to the
minimal interference from topographical variations and external factors.

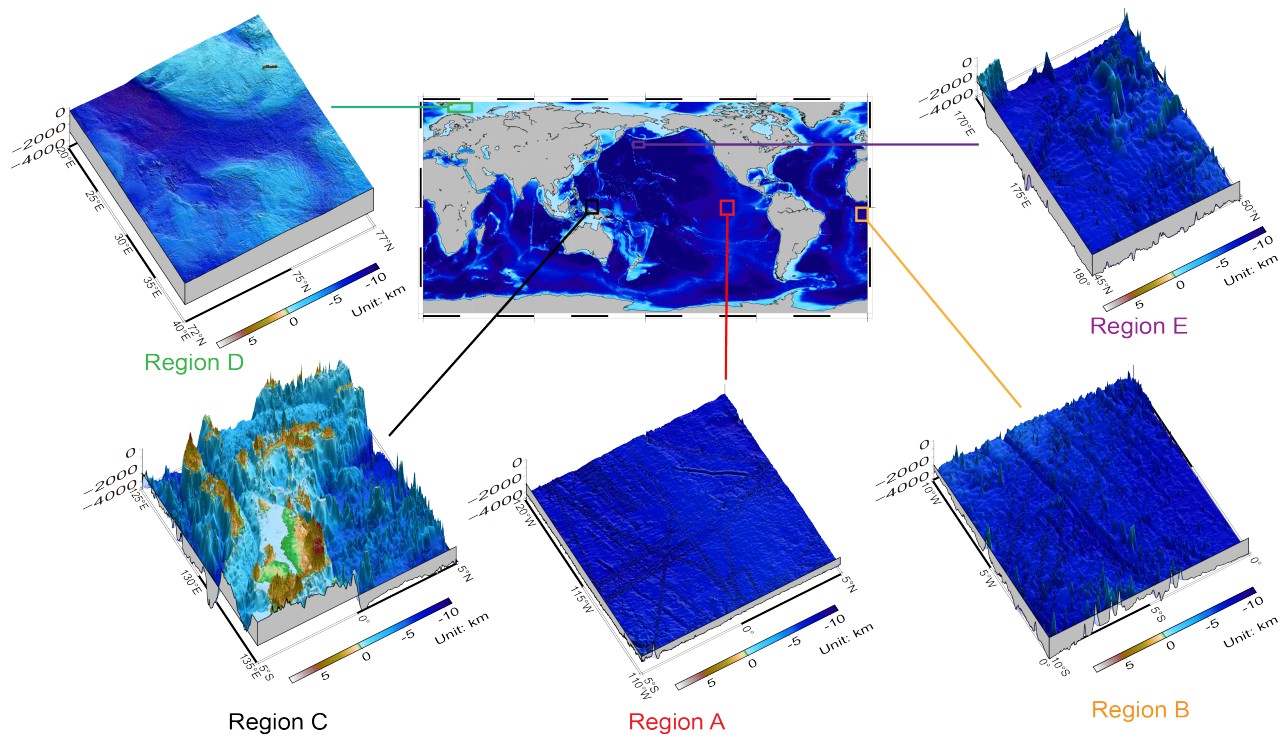

**Figure 9.** Topographic representation of five selected regions for model difference analysis: (A) the open deep-sea area of the South Pacific far from land, (B) the southeastern Atlantic region near the continental shelf with varied topography, (C) the area near the Indonesian archipelago with complex seafloor topography, (D) the high-latitude North Atlantic region near the Arctic Circle, and (E) the western Pacific deep-sea region characterized by complex underwater terrain but well-covered by satellites.

**Table 7.** Statistical comparison of differences obtained by subtracting the VGGA model from the SIO V32.1 curv model in different regions (unit: E)

| Regions | Lat and lon | Mean Depth (m) | Max | Min | Mean | STD | RMS |
|---------|-------------|----------------|-----|-----|------|-----|-----|
| A | $-5° \sim 5°, -120° \sim -110°$ | -4077.64 | 25.64 | -22.27 | -0.05 | 4.20 | 4.20 |
| B | $-10° \sim 0°, -10° \sim 0°$ | -4476.59 | 58.73 | -28.32 | -0.04 | 4.22 | 4.22 |
| C | $-5° \sim 5°, 125° \sim 135°$ | -2585.25 | 370.86 | -382.23 | -0.76 | 17.59 | 17.61 |
| D | $72° \sim 77°, 20° \sim 40°$ | -257.23 | 45.18 | -39.44 | -0.06 | 3.81 | 3.81 |
| E | $45° \sim 50°, 170° \sim 180°$ | -5444.67 | 29.37 | -24.31 | -0.03 | 4.02 | 4.02 |

In contrast, Region B in the southeastern Atlantic (10° S to 0°, 10° W to 0°) features varying seafloor topography with an average depth of approximately -4487 meters and is closer to the continental shelf. The seafloor topography here is more complex than in Region A. As a result, the residual extremes between the VGGA and SIO V32.1 curv models are significantly larger, with maximum and minimum differences of 58.73 E and -28.32 E, respectively. This indicates that the increased



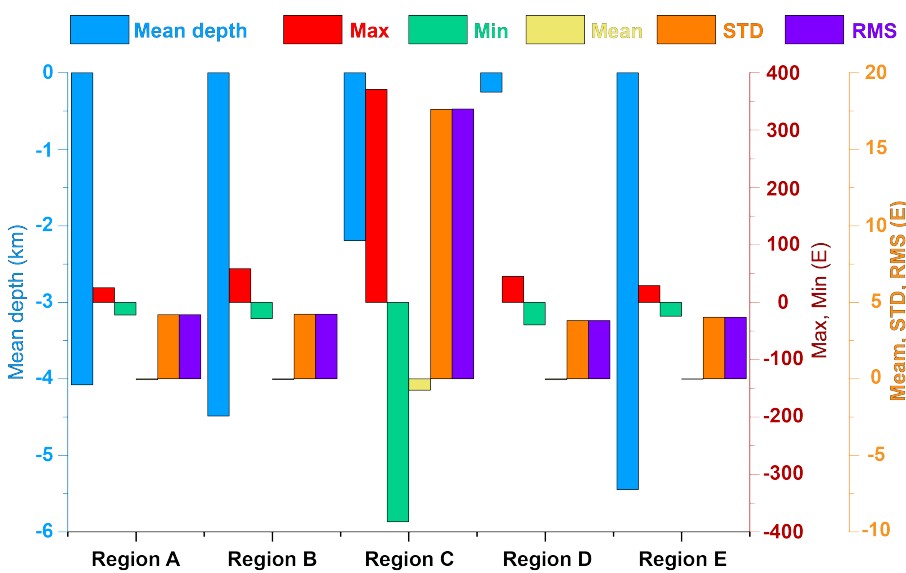

**Figure 10.** Comparison of differences between the VGGA model and the SIO V32.1 curv model across selected regions. The regions include: (A) a deep-sea area in the South Pacific, (B) a southeastern Atlantic region with varied topography, (C) the Indonesian archipelago with complex seafloor features, (D) the high-latitude North Atlantic near the Arctic Circle, and (E) a western Pacific deep-sea area. The bars represent key metrics such as mean depth, maximum and minimum differences, mean, STD, and RMS of the residuals.

complexity of the seafloor topography introduces greater challenges in the inversion process, resulting in larger discrepancies between the models. These results suggest that in regions with more intricate terrain, satellite altimetry data quality is affected by topographical complexity, resulting in increased uncertainty and decreased model consistency.

Compared to Regions A and B, the significantly higher residual values in Region C (5° S to 0°, 125° E to 135° E) indicate that the complexity of the seafloor topography, combined with the multipath effect, poses greater challenges for the DTU21 MSS

model in low-latitude areas. The steep slopes and varied elevations associated with seamounts and deep trenches introduce greater variability in gravity anomaly signals, complicating the inversion process and decreasing model consistency. As a result, the large residual extremes observed between the VGGA and SIO V32.1 models suggest that satellite altimetry data accuracy in this region is more susceptible to interference from local topographical features. These findings highlight that in low-latitude regions with highly complex terrain structures, the model performance is less reliable compared to relatively flat

or less complex regions, as seen in Regions A and B.

Region D in the North Atlantic near the Arctic Circle (72° N to 77° N, 20° E to 40° E) has a relatively smooth seafloor topography with an average shallow depth of approximately -256 meters. Despite its high-latitude location, the region lacks significant topographical undulations or complex structures. These flat conditions result in smaller residuals between the VGGA and SIO V32.1 models, with maximum and minimum differences of 45.18 E and -39.44 E, respectively, and an STD and RMS

of 3.81 E. The strong consistency between the models in this region is likely due to stable ocean currents in polar areas and comprehensive satellite coverage.





Region E in the western Pacific deep-sea area (45° N to 50° N, 170° E to 180° E) is characterized by an extremely deep average depth of about -5446 meters. Although the seafloor topography here is relatively complex compared to Region B, Region E benefits from better satellite coverage due to its mid-latitude location. Consequently, the residuals between the VGGA and SIO V32.1 models are relatively small, with maximum and minimum differences of 29.37 E and -24.31 E, respectively. The STD and RMS are 4.02 E, indicating better model performance in this region compared to Region B.

The comparison of these five regions confirms that the DTU21 MSS model effectively manages complex ocean dynamics in deep-sea and high-latitude areas, including regions near polar ice caps, demonstrating its reliability across global oceanic environments. The results indicate that the consistency between the inversion results and the SIO V32.1 model is not directly correlated with sea depth or latitude. Instead, it is the seafloor topography, particularly in shallow-water regions and areas with complex underwater terrain, that leads to instability in satellite altimetry data, which directly affects the consistency between the two models.

Due to the lack of in-situ VGGA measurements, this study aims to explore alternative methods to validate the performance of the SDUST2023VGGA model (hereafter referred to as SDUSTVGGA). Given that both the SIO V32.1 Curv model and the VGGA model are derived from satellite altimetry data, they can partially verify the methodology but not the accuracy of the inversion results. Therefore, in this exploratory attempt, the GEBCO bathymetric model was used as an alternative reference to assess the SDUSTVGGA model. However, the indirect and weak relationship between bathymetric features and VGGAs has led to relatively low correlation coefficients and R-squared values, limiting the strength of the conclusions drawn from this analysis. Consequently, these results should be interpreted with caution, as the experiment's limitations suggest that the current approach may not fully capture the model's accuracy.

The analysis results are illustrated in Figure 11, which shows the correlation coefficients and model performance metrics across different evaluation methods, highlighting the limitations inherent in using bathymetric features as proxies for VGGA validation.

Spearman rank correlation analysis revealed weak positive monotonic relationships between the VGGAs predicted by both models and the GEBCO bathymetric data. Kendall's tau showed weak positive relationships, with values of 0.0634 for V32.1 and 0.0696 for VGGA. Pearson correlation coefficients were slightly higher, at 0.1179 for V32.1 and 0.1361 for VGGA. These results suggest that while the models exhibit some degree of correlation with the bathymetric data, the relationships remain weak, reflecting the models' limited ability to capture the complexity of the underlying geophysical processes.

Both models performed poorly in linear regression, with R-squared values of 0.0139 for V32.1 and 0.0185 for VGGA. These low R-squared values underscore the inability of simple linear models to adequately explain the variance in the GEBCO bathymetric data, highlighting the need for more sophisticated, non-linear modeling approaches. Polynomial regression models led to modest improvements in predictive performance, with R-squared values of 0.0221 for V32.1 and 0.0230 for VGGA. While these results suggest that non-linear models can better capture some underlying data patterns, the improvements were still relatively minor, indicating that further refinement is needed.

Support vector regression (SVR) was employed to explore the models' capacity to capture non-linear relationships. Initial results from basic SVR models revealed substantial improvements over linear and polynomial regression, with R-squared





**Figure 11.** Correlation and model performance comparisons for different bathymetric and satellite-derived models. Top: Correlation coefficients (Pearson, Spearman, Kendall) between the GEBCO bathymetric model (A) and the SIO V32.1 Curv model (B), as well as between the GEBCO model (A) and the VGGA model (C), illustrating the weak associations between these datasets. Bottom: $R^2$ values from various regression models (Linear Regression, Polynomial Regression, SVR, and MLP) applied to compare GEBCO (A) with V32.1 Curv (B), and GEBCO (A) with VGGA (C), demonstrating the impact of non-linear modeling and hyperparameter optimization on improving model performance.

values of 0.3619 for V32.1 and 0.3745 for VGGA. To further optimize model performance, both grid search and random search techniques were applied. These efforts resulted in higher R-squared values: 0.4228 for VGGA and 0.4093 for V32.1



using grid search, and 0.4123 for VGGA and 0.4027 for V32.1 using random search. These improvements underscore the
importance of hyperparameter tuning in enhancing model performance.

Neural network models were also explored to capture more intricate non-linear relationships. A shallow multi-layer perceptron (MLP) model, consisting of two hidden layers with 50 units each, demonstrated further improvement over the SVR models, achieving R-squared values of 0.4454 for V32.1 and 0.4599 for VGGA. These results surpassed all previous models, suggesting that the shallow MLP was effective at learning complex patterns in the data.

However, hyperparameter optimization of the MLP models through grid search and random search did not yield significant improvements. Grid search optimization for MLP models with two hidden layers (50 units each) resulted in R-squared values of 0.4450 for V32.1 and 0.4585 for VGGA, while random search optimization produced similar results, with R-squared values of 0.4453 for V32.1 and 0.4568 for VGGA. These findings indicate that the MLP models' performance did not improve substantially with hyperparameter optimization, suggesting that further tuning or alternative model architectures may be
needed.

When deeper neural networks were tested, consisting of three hidden layers with 100 units each, the performance decreased significantly. The deep MLP models achieved R-squared values of 0.0376 for V32.1 and 0.0416 for VGGA, indicating that the added complexity led to overfitting without providing additional predictive power. These findings suggest that simpler architectures, such as the shallow MLP, are more suitable for this problem as they balance model complexity with predictive
accuracy.

In conclusion, the findings suggest that while deeper neural networks may improve performance in some cases, the shallow MLP model provided the optimal balance between complexity and accuracy for this task. The results emphasize the effectiveness of non-linear models, particularly SVR and MLP, in capturing the complex relationships between VGGAs and bathymetric data. While each successive model showed some improvement, the limitations of the current experimental setup must be ac-
knowledged. The weak correlations and relatively low R-squared values indicate that relying solely on VGGA is insufficient due to its indirect relationship with bathymetric features. Introducing additional data might improve correlations or model performance but could also introduce more error, complicating interpretation and reducing robustness. Further refinement and exploration of more advanced techniques are necessary to better capture the complexities inherent in geophysical data.

## 6 Conclusions

This study used the gridded DTU21 MSS model, combined with multi-directional MSS data, to develop the global VGGA model, SDUST2023VGGA. The reliability of the SDUST2023VGGA model was validated by comparing it with the SIO V32.1 curv model. The comparison confirmed the effectiveness of the proposed method, which considers the complex oceanic environment and effectively suppresses the amplification of geoid uncertainties caused by second-order derivative calculations.

The process starts by dividing the global oceanic area into multiple sub-regions to address computational limitations. Next,
DTU21 MSS data from multiple directions are used, and the influence of CNES-CLS22 MDT is subtracted to derive the geoid. The remove-restore method is then applied to eliminate the long-wavelength signals of the gravity field, obtaining the residual



geoid. In the third step, the weighted least squares method is employed to calculate the residual VGGA based on the residual geoid, after which the long-wavelength signals are restored to construct the VGGA model within each sub-region. Finally, all sub-regions are merged to construct the SDUST2023VGGA model (hereafter referred to as VGGA), covering the latitude range from 80° S to 80° N.

The VGGA model was compared with the SIO V32.1 curv model, both derived from satellite altimetry data. The following conclusion can be drawn: On a global scale, the VGGA model, derived through multi-directional sea surface height data inversion, shows a high degree of consistency with the SIO V32.1 model. In the longitudinal direction, the differences exhibit significant numerical fluctuations without a consistent trend, indicating that the VGGA in this direction is affected by various geographical and geophysical factors. In contrast, in the latitudinal direction, the complexity of topography in high-latitude regions leads to greater discrepancies between the VGGA and SIO V32.1 models compared to those in mid- and low-latitude regions.

The inversion results are directly affected by coastal proximity and topographical features. In open ocean basins far from the shore with relatively flat terrain, there is a strong consistency between the VGGA and SIO V32.1 models. However, in regions near complex coastal areas, the discrepancies between the models are more pronounced. This is likely due to the influence on satellite altimetry data quality and differences in processing strategies. Indirectly, sea depth also plays a role; in shallow waters, which often correspond to complex coastlines, the consistency between the two models is poor. Conversely, in deep-sea regions where the terrain is generally flat, the consistency is significantly better.

## 7  Data availability

The datasets used in this study are listed in Table 8, along with their corresponding DOIs and access information.

**Table 8.** Data availability and DOI information

| Dataset | DOI |
| --- | --- |
| DTU21MSS | https://doi.org/ |
| | 10.11583/DTU.19383221.v1 (Andersen et al., 2023) |
| CNES-CLS22MDT | https://doi.org/ |
| | 10.22541/essoar.170158328.85804859/v1 (Jousset et al., 2023) |
| XGM2019e | https://doi.org/ |
| | 10.5880/ICGEM.2019.007 (Zingerle et al., 2020) |
| V32.1 | https://doi.org/ |
| | 10.1126/science.1258213 (Sandwell et al., 2014; Garcia et al., 2014) |
| GEBCO2024 | https://doi.org/ |
| | 10.5285/1c44ce99-0a0d-5f4f-e063-7086abc0ea0f (GEBCO Bathymetric Compilation Group 2024, 2024) |

Calculations in this study utilized the ICGEM service (Ince et al., 2019), which was used to compute the long-wavelength component of the vertical gradient of gravity anomaly (VGGA). The service can be accessed at https://icgem.gfz-potsdam.de/tom_longtime.

The global ocean vertical gradient of gravity anomaly dataset, SDUST2023VGGA, is available at the Zenodo repository
(https://doi.org/10.5281/zenodo.14177000 (Zhou et al., 2024)). This dataset includes global ocean VGGA in NetCDF format (latitude vector, longitude vector, and VGGA matrix).

*Author contributions.* Ruichen Zhou and Jinyun Guo contributed equally to the conceptualization and design of the study. Ruichen Zhou conducted the primary investigation, data curation, formal analysis, and wrote the original draft of the manuscript. Jinyun Guo supervised the project, provided resources, and contributed significantly to the methodology and manuscript revision. Shaoshuai Ya was involved in data
acquisition and supported data analysis. Heping Sun offered technical support, guidance, and contributed to software development. Xin Liu performed validation and contributed to the review and editing of the manuscript. All authors have read and approved the final version of the manuscript.

*Competing interests.* The authors declare that they have no conflict of interest.

*Disclaimer.* The views expressed in this article are those of the authors and do not necessarily reflect the views of their affiliated institutions.

*Acknowledgements.* The authors express their appreciation to DTU (Technical University of Denmark), CNES (Centre National d'Études Spatiales), TUM (Technical University of Munich), SIO (Scripps Institution of Oceanography), and GEBCO (General Bathymetric Chart of the Oceans) for their commitment to open data policies. The availability of high-quality, accessible datasets from these institutions significantly advances geosciences, enabling researchers worldwide to conduct robust and innovative studies. We also thank the International Centre for Global Earth Models (ICGEM) service team for their efforts in providing and maintaining global gravity field models, which
greatly facilitated this research.

Additionally, we acknowledge the contributions of all cited authors. Figures in this study were generated using the Generic Mapping Tools (Wessel et al., 2019), which greatly facilitated the visualization and interpretation of results. GMT was developed by Pål Wessel, Professor of Geophysics at the University of Hawai'i at Mānoa, and his colleagues. We extend our deepest gratitude to the late Professor Wessel for his invaluable contributions to the scientific community, particularly in the field of geosciences.
This study was partially supported by the National Natural Science Foundation of China (grant Nos. 42430101, 42192535, and 42274006). We are grateful for the valuable feedback provided by the reviewers.





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
