# Peer review of "SDUST2023VGGA: A Global Ocean Vertical Gradient of Gravity Anomaly Model Determined from Multidirectional Data from Mean Sea Surface"

_Earth System Science Data, 2024_

## Referee Comment (RC1)

**General Comments:**

This manuscript presents a vertical gravity gradient anomaly (VGGA) model constructed based on the DTU21 mean sea surface model. The research is thorough and meets quality standards. This makes me willing to apply this model in my future work. However, several issues need to be addressed before publication, particularly in terms of organization and clarity. These improvements would enhance the paper's readability and impact.
* * *
**Major Comments:**

1. **P1, l10:** Replace "actual ocean environment" with "real ocean environment."

2. **P1, l12-13:** Rephrase for clarity: "The comparison between the VGGA and the SIO V32.1 model shows a residual mean of -0.08 Eötvös (E) and an RMS of 8.50 E, demonstrating high consistency on a global scale."

3. **P1, l16:** Consider using "multidirectional" instead of "multi-directional" for a more modern and streamlined expression.

4. **P2, l29:** Specify that the paper used $V_{XX}$, $V_{YY}$, and $V_{ZZ}$.

5. **P2, l44:** Maintain consistency in terminology. Since "gradients of gravity" was used earlier, ensure it is used consistently instead of "gravity gradients."

6. **P3, l86:** Replace "mean sea surface" with its abbreviation MSS after first defining it. Refer to the https://www.earth-system-science-data.net/submission.html#reference for consistency.

7. **P4, l118:** Clarify that only the "curv" dataset of SIO V32.1 was used, rather than the entire dataset. Update Figure 4 accordingly.

8. **P5, l128:** The model's name, SDUST2023VGGA, is acceptable; however, inconsistent use of VGGA throughout the text creates confusion. Ensure a clear distinction is made.

9. **P6, Fig 1:** While Arial font is acceptable, the latitude/longitude labels and partition indices lack clarity and should be improved. For Fig 2, the title should include "example," as seen in Fig 3, to account for the partition changes beyond 60° latitude.

10. **P6, l155:** Resolve the inconsistency between VGGA and SDUST2023VGGA to avoid reader confusion.

11. **P7, l161:** Address grammatical issues in this line.

12. **P7, l175:** Reorganize the language to improve clarity and readability.

13. **P7, l184:** Use "geoid" instead of "geoid structure," and enhance the explanation of Equation (6).

14. **P8, Fig 3:** The use of bold Arial font enhances visibility. Apply this style to Fig 1 for consistency.

15. **P8, l186:** Replace "multiple directions" with "multidirectional," and revise Equation (10) to align with the weighted least squares method described earlier.

16. **P8, l189:** Consider replacing "formula" with "equation," as it is more commonly used in technical writing.

17. **P8, l198:** The formatting of equations is acceptable and aligns with journal requirements.

18. **P9, l203:** Replace "directions" with "components" for technical accuracy.

19. **P9, l207:** Ensure matrices are consistently printed in boldface.

20. **P10, Fig 4:** The handling of overlapping partitions is explained in Fig 4 but is not adequately addressed in the text. Add a clear explanation.

21. **P9, l217:** State the degree and order of XGM2019e used in model construction.

22. **P9, l218:** There seems to be an extraneous ")" that might be a typographical error. Additionally, the term "estimated" could be replaced with a more precise term. Reorganizing this paragraph would enhance readability and clarity.

23. **P12–13:** While the analysis is comprehensive, some referenced articles appear tangential to the main findings and might not directly support the conclusions. Consider removing or replacing them with more directly relevant studies. Additionally, ensure consistent usage of latitude and longitude references throughout.

24. **P14, l305:** The term "additional" might not accurately describe the filtering process. It would be clearer to specify that the three-sigma criterion filters 1.41% of the data.

25. **P15, Fig 7:** Correct discrepancies in labels and data in the top-right corner of the figure.

26. **P15, l330:** Remove repeated text caused by potential copy-paste errors.

27. **P18, l354:** Simplify "residual values" to "residuals" per the https://www.earth-system-science-data.net/submission.html.

28. **P19, l379:** Correct noun errors in this line.

29. **P19, l389:** Merge "The modeling results in different regions are shown in Figure 10." with

the previous paragraph for better flow.

30. **P20, Fig 9:** While the color scheme is visually appealing, bolden the text to improve clarity.

31. **P21, Fig 10:** Thicken the axis lines for better readability.

32. **P22, I428:** to improve the organization and readability of the manuscript, I recommend that this section be moved to a separate subsection under Section 5. Additionally, while the results presented here reflect an initial attempt by the authors, and the manuscript appropriately acknowledges their limitations, these results still offer some level of validation for the accuracy of the model.

33. **P22, I429:** Use the format "hereafter referred to as SDUSTVGGA" consistently throughout the text for clarity.

34. **P23, Fig 11:** Consider splitting this figure into two for better presentation.

35. **P24, I480:** Rewrite the conclusion to remove redundant expressions and incorporate specific details from the results and analysis.

36. **P28, I594:** The citation of Kim et al. does not convincingly support the manuscript's findings and should be removed.

---

## Author Response (AR1)

**General Response to Reviewers' Comments**

Dear Editor and Reviewers,

We sincerely thank you for your valuable feedback and constructive suggestions on our manuscript. Your detailed and insightful comments have greatly contributed to improving the clarity, accuracy, and overall quality of our work. We particularly appreciate the thoroughness and professionalism of your reviews, which have helped us address key issues and refine our methodology and presentation.

In response to your comments, we have revised the manuscript to address all the specific points raised in the reviews. Additionally, we conducted further experiments to strengthen the study and support our conclusions. To enhance the readability and flow of the manuscript, we have also optimized the language and structure, carefully polishing the text to ensure that the ideas are presented clearly and concisely. These improvements aim to make the manuscript more accessible to a broader audience.

We have carefully addressed all the comments and made corresponding revisions to the manuscript. Below, we provide a detailed point-by-point response to each comment, and all changes have been highlighted in the revised manuscript for ease of reference.

**Reviewer 1**

1. **P1, l10**: Replace "actual ocean environment" with "real ocean environment."
   **Response:** We have replaced "actual ocean environment" with "real ocean environment" to better reflect the terminology commonly used in geophysical studies. This change improves the clarity and consistency of the manuscript.
   Changes in manuscript: Line 10, Page 1.

2. **P1, l12-13**: Rephrase for clarity: "The comparison between the VGGA and the SIO V32.1 model shows a residual mean of -0.08 Eötvös (E) and an RMS of 8.50 E, demonstrating high consistency on a global scale."
   **Response:** The sentence has been rephrased for clarity and improved readability. The updated version maintains the original meaning while enhancing the flow of the text.
   Changes in manuscript: Lines 12-13, Page 1.

3. **P1, l16**: Consider using "multidirectional" instead of "multi-directional" for a more modern and streamlined expression.
   **Response:** We have updated "multi-directional" to "multidirectional," as this is the more

modern and concise form, in line with current scientific writing conventions. This change ensures consistency with the terminology used in the field.

Changes in manuscript: Line 16, Page 1.

4.  **P2, l29**: Specify that the paper used $V_{XX}$, $V_{YY}$, and $V_{ZZ}$.

    **Response:** We have clarified that the study specifically utilized the gradients of gravity from the GOCE satellite, including the components $V_{XX}$, $V_{YY}$, and $V_{ZZ}$. This addition provides more specific details about the data and avoids ambiguity.

    Changes in manuscript: Line 29, Page 2.

5.  **P2, l44**: Maintain consistency in terminology. Since "gradients of gravity" was used earlier, ensure it is used consistently instead of "gravity gradients."

    **Response:** We have defined "mean sea surface" (MSS) at first mention and used the abbreviation consistently thereafter. This change adheres to the recommended formatting for clarity and consistency.

    Changes in manuscript: Line 86, Page 3.

6.  **P3, l86**: Replace "mean sea surface" with its abbreviation MSS after first defining it. Refer to the https://www.earth-system-science-data.net/submission.html#reference for consistency.

    **Response:** We have defined "mean sea surface" (MSS) at first mention and used the abbreviation consistently thereafter. This change adheres to the recommended formatting for clarity and consistency.

    Changes in manuscript: The term has been consistently replaced across the entire manuscript.

7.  **P4, l118**: Clarify that only the "curv" dataset of SIO V32.1 was used, rather than the entire dataset. Update Figure 4 accordingly.

    **Response:** We have clarified that only the "curv" dataset of the SIO V32.1 model was used in the analysis. Additionally, we have updated Figure 4 to accurately reflect this change.

    Changes in manuscript: Line 118, Page 4; Figure 4.

8.  **P5, l128**: The model's name, SDUST2023VGGA, is acceptable; however, inconsistent use of VGGA throughout the text creates confusion. Ensure a clear distinction is made.

    **Response:** We have revised the manuscript to use the model name "SDUSTVGGA" consistently throughout, ensuring clarity and avoiding any confusion between different references to the model.

    Changes in manuscript: Line 128, Page 5.

9.  **P6, Fig 1**: While Arial font is acceptable, the latitude/longitude labels and partition indices

lack clarity and should be improved. For Fig 2, the title should include "example," as seen in Fig 3, to account for the partition changes beyond 60° latitude.

**Response:** We have improved the clarity of the latitude/longitude labels and partition indices in Figure 1 for better readability. We have also updated the title of Figure 2 to include the word "example," aligning it with Figure 3, to reflect the partition changes correctly beyond 60° latitude.

Changes in manuscript: Figure 1

10. **P6, l155**: Resolve the inconsistency between VGGA and SDUST2023VGGA to avoid reader confusion.

    **Response:** Inconsistencies between "VGGA" and "SDUST2023VGGA" have been resolved, and "SDUSTVGGA" is now used consistently throughout the manuscript. This change ensures clarity and avoids confusion for readers.

    Changes in manuscript: Line 155, Page 6.

11. **P7, l161**: Address grammatical issues in this line.

    **Response:** The grammatical issues in the specified line have been corrected to enhance readability and ensure accurate communication of the intended meaning.

    Changes in manuscript: Line 161, Page 7.

12. **P7, l175**: Reorganize the language to improve clarity and readability.

    **Response:** The language has been reorganized to improve both clarity and readability, ensuring a more coherent presentation of the results.

    Changes in manuscript: Line 175, Page 7.

    **P7, l184**: Use "geoid" instead of "geoid structure," and enhance the explanation of Equation (6).

    Changes in manuscript: Line 184, Page 7; Equation (6).

    **Response:** The term "geoid structure" has been replaced with "geoid" to align with standard geophysical terminology. Additionally, we have expanded the explanation of Equation (6), providing more detailed descriptions of the symbols and wavelengths involved.

14. **P8, Fig 3**: The use of bold Arial font enhances visibility. Apply this style to Fig 1 for consistency.

    **Response:** We have applied the bold Arial font style to the labels and text in Figure 1 to ensure consistency with the style used in Figure 3, improving the overall visual clarity.

    Changes in manuscript: Figure 1.

15. **P8, l186**: Replace "multiple directions" with "multidirectional," and revise Equation (10) to align with the weighted least squares method described earlier.

**Response:** The phrase "multiple directions" has been replaced with "multidirectional," as it is more precise. We have also revised Equation (10) to ensure it aligns with the weighted least squares method described earlier in the manuscript.

Changes in manuscript: Line 186, Page 8.

16. **P8, l189**: Consider replacing "formula" with "equation," as it is more commonly used in technical writing.

    **Response:** "Formula" has been replaced with "equation" to adhere to more conventional technical writing terminology, which enhances the manuscript's consistency.

    Changes in manuscript: Line 189, Page 8.

17. **P8, l198**: The formatting of equations is acceptable and aligns with journal requirements.

    **Response:** No changes were needed, as the formatting of the equations already complies with the journal's requirements.

    Changes in manuscript: Line 198, Page 8.

18. **P9, l203**: Replace "directions" with "components" for technical accuracy.

    **Response:** "Directions" has been replaced with "components" to reflect more accurate technical language in line with the terminology used in the field.

    Changes in manuscript: Line 203, Page 9.

19. **P9, l207**: Ensure matrices are consistently printed in boldface.

    **Response:** All matrices have been reviewed and consistently formatted in boldface throughout the manuscript, as per the journal's guidelines.

    Changes in manuscript: Line 207, Page 9.

20. **P10, Fig 4**: The handling of overlapping partitions is explained in Fig 4 but is not adequately addressed in the text. Add a clear explanation.

    **Response:** We have added a detailed explanation in the manuscript regarding the handling of overlapping partitions, clarifying the trimming and merging processes. Figure 4 has been updated to better illustrate these steps.

    Changes in manuscript: Page 10; Figure 4.

21. **P9, l217**: State the degree and order of XGM2019e used in model construction.

    **Response:** We have clearly stated that the XGM2019e model was used up to degree and order 2190, providing more transparency on the model's configuration. This addition ensures that readers have a clear understanding of the model's scope and limitations.

    Changes in manuscript: Line 217, Page 9.

22. **P9, l218**: There seems to be an extraneous ")" that might be a typographical error. Additionally, the term "estimated" could be replaced with a more precise term.

Reorganizing this paragraph would enhance readability and clarity.

**Response:** The typographical error has been corrected, and the term "estimated" has been replaced with more precise language. The paragraph has also been reorganized for better readability and clearer communication. These changes improve the overall flow and accuracy of the text.

Changes in manuscript: Line 218, Page 9.

23. **P12–13**: While the analysis is comprehensive, some referenced articles appear tangential to the main findings and might not directly support the conclusions. Consider removing or replacing them with more directly relevant studies. Additionally, ensure consistent usage of latitude and longitude references throughout.

    **Response:** We have reviewed and ensured the consistent use of latitude/longitude throughout the manuscript, removing any tangential references that could confuse the reader. This change enhances the clarity and consistency of the manuscript.

    Changes in manuscript: Pages 12-13.

24. **P14, l305**: The term "additional" might not accurately describe the filtering process. It would be clearer to specify that the three-sigma criterion filters 1.41% of the data.

    **Response:** The description of the filtering process has been clarified to specify that the three-sigma criterion filters out 1.41% of the data. This change provides a more precise and accurate description of the methodology.

    Changes in manuscript: Line 305, Page 14.

25. **P15, Fig 7**: Correct discrepancies in labels and data in the top-right corner of the figure.

    **Response:** Discrepancies in labels and data in the top-right corner of Figure 7 have been corrected to match the corresponding data in the manuscript. This ensures the accuracy and reliability of the presented results.

    Changes in manuscript: Figure 7.

26. **P15, l330**: Remove repeated text caused by potential copy-paste errors.

    **Response:** Repeated text has been removed to streamline the manuscript and ensure that the content flows logically without redundancy. This change improves the readability and coherence of the text.

    Changes in manuscript: Line 330, Page 15.

27. **P18, l354**: Simplify "residual values" to "residuals" per the https://www.earth-system-science-data.net/submission.html.

    **Response:** "Residual values" has been simplified to "residuals," in accordance with the journal's submission guidelines.

Changes in manuscript: Line 354, Page 18.

28. **P19, l379**: Correct noun errors in this line.

    **Response:** Noun errors in the specified line have been corrected to ensure clarity and proper grammatical structure. This change enhances the readability and accuracy of the text.

    Changes in manuscript: Line 379, Page 19..

29. **P19, l389**: Merge "The modeling results in different regions are shown in Figure 10." with the previous paragraph for better flow.

    **Response:** The sentence has been merged with the preceding paragraph to improve the flow of the text and create a more cohesive structure. This change enhances the overall readability of the manuscript.

    Changes in manuscript: Line 389, Page 19.

30. **P20, Fig 9**: While the color scheme is visually appealing, bolden the text to improve clarity.

    **Response:** The text in Figure 9 has been boldened to enhance its visibility and readability. This change ensures that the figure is clearer and easier to interpret, particularly for readers who may view the manuscript in low-resolution formats.

    Changes in manuscript: Figure 9.

31. **P21, Fig 10**: Thicken the axis lines for better readability.

    **Response:** The axis lines in Figure 10 have been thickened to improve readability, ensuring that the figure is clearer and easier to interpret. This adjustment enhances the visual presentation of the data and aligns with best practices for scientific figures.

    Changes in manuscript: Figure 10.

32. **P22, Section 5**: To improve the organization and readability of the manuscript, I recommend that this section be moved to a separate subsection under Section 5. Additionally, while the results presented here reflect an initial attempt by the authors, and the manuscript appropriately acknowledges their limitations, these results still offer some level of validation for the accuracy of the model.

    **Response:** A new subsection under Section 5 has been created to improve organization and readability. This restructuring ensures that the results are presented in a more logical and accessible manner, while maintaining the manuscript's acknowledgment of the limitations and the validation provided by the results.

    Changes in manuscript: Section 5 has been restructured, and a new subsection has been added.

33. **P22, l429**: Use the format "hereafter referred to as SDUSTVGGA" consistently throughout the text for clarity.

**Response:** The format "hereafter referred to as SDUSTVGGA" has been applied consistently throughout the manuscript for clarity and to avoid ambiguity. This change ensures that readers can easily follow the references to the model without confusion.

Changes in manuscript: Line 429, Page 22, and throughout the manuscript.

34. **P23, Fig 11**: Consider splitting this figure into two for better presentation.

**Response:** Figure 11 has been split into two separate figures to improve clarity and presentation. This change allows for a more detailed and focused display of the data, making it easier for readers to interpret the results.

Changes in manuscript: Figure 11 has been replaced with two new figures

35. **P24, l480**: Rewrite the conclusion to remove redundant expressions and incorporate specific details from the results and analysis.

**Response:** The conclusion has been rewritten to eliminate redundancy and incorporate more specific details drawn from the results and analysis. This revision ensures a more concise and impactful summary that highlights the key findings and their implications.

Changes in manuscript: Line 480, Page 24; Conclusion section.

36. **P28, l594**: The citation of Kim et al. does not convincingly support the manuscript's findings and should be removed.

**Response:** The citation of Kim et al. has been removed as it was not directly relevant to the manuscript's findings. This change improves the focus and relevance of the references, ensuring that only the most pertinent literature is cited.

Changes in manuscript: Line 594, Page 28; References section.

**Reviewer 2**

1. P1L8: The order of the gravity field model should be clearly specified. Additionally, the term "original geoid" is used incorrectly and should be corrected for accuracy.

   **Response:** We have clarified the degree and order of the gravity field model in the manuscript and corrected "original geoid" to "full-wavelength geoid" to ensure accuracy. This change aligns with standard geophysical terminology and improves the precision of the manuscript.

   Changes in manuscript: Line 8, Page 1.

2. P1L18: The term "SDUST2023VGGA model" should be used consistently throughout the text instead of alternating with "dataset" to avoid confusion.

   **Response:** We have ensured consistent use of the term "SDUST2023VGGA model" throughout the manuscript and replaced all instances of "dataset" to prevent confusion. This change enhances the clarity and consistency of the manuscript.

Changes in manuscript: Line 18, Page 1, and throughout the manuscript.

3. P1L20: FYI, the SWOT-derived VGG has been released by SIO using wide-swath data, the first evaluation suggested that this dataset was much better than that derived solely from nadir altimetry (1 year SWOT was better than 30 years of nadir altimetry). The authors may include the SWOT information in deriving the global VGG.

**Response:** Thank you for highlighting the potential of SWOT-derived VGG data. In our study, we utilized multi-year mean sea level (MSS) data for the following reasons:

a. The currently available SWOT data spans approximately one year, which is insufficient for developing multi-year averages required for stable long-term trends.

b. Multi-year MSS data helps reduce the influence of transient ocean phenomena, leading to a more reliable VGGA model.

c. SWOT data lacks directly usable MSS models, limiting its immediate applicability to our methodology.

We recognize SWOT's value and plan to incorporate it in future studies to refine and validate the VGGA model further.

Changes in manuscript: Line 20, Page 1; Discussion section.

4. P3L55-L64: The language in this section should be restructured to enhance clarity and readability.

**Response:** We have restructured this section to improve clarity and enhance readability. The revised version ensures a more coherent and accessible presentation of the methodology.

Changes in manuscript: Lines 55-64, Page 3.

5. P3L64: Why not use observed SSHs from multi-satellite missions, why used an existing MSS model for deriving the VGG? The authors may include the possible reasons.

**Response:** Thank you for this thoughtful question. We chose to use the DTU21MSS model instead of directly using observed Sea Surface Heights (SSH) from multiple satellite missions for the following reasons:

a. The DTU21MSS model provides long-term stable sea level data, optimized and validated over multiple years, ensuring consistent trends. In contrast, multi-satellite SSH data may introduce inconsistencies due to differences in satellite missions and processing methods.

b. The DTU21MSS model effectively mitigates oceanic phenomena (e.g., tides, wind waves), ensuring reliable data. Using raw multi-satellite SSH data would require additional processing, which is not yet fully integrated into our current methodology.

c. Integrating SSH data from various satellites involves complex calibration to ensure consistency and accuracy, whereas the DTU21MSS model offers a validated, streamlined data

pipeline that meets our requirements.

By using the DTU21MSS model, we ensure that the VGGA derivation is based on reliable and well-processed data, crucial for capturing long-term sea level trends.

Changes in manuscript: Line 64, Page 3; Methodology section.

6. P3L65: It appears that only the vertical gravity gradient anomaly has been modeled. Consequently, the study's objectives should be revised to accurately reflect this specific focus.

**Response:** Thank you for your observation. We have revised the study's objectives to explicitly state that the focus is on modeling the vertical gravity gradient anomaly, aligning the objectives with the study's scope.

Changes in manuscript: Line 65, Page 3.

7. P3L86: There are multiple instances of inconsistent abbreviation usage throughout the manuscript. Please ensure that all abbreviations are defined upon their first occurrence and maintained consistently throughout the text.

**Response:** All abbreviations have been defined at their first occurrence, and we have ensured consistency in their usage throughout the manuscript. This change improves the clarity and readability of the text.

Changes in manuscript: Line 86, Page 3, and throughout the manuscript.

8. P4: While the CNES-CLS22 MDT is mentioned as having increased resolution, the original resolution of the model is not specified and should be included. Additionally, the order of the XGM2019e model is not mentioned and should be provided for completeness.

**Response:** Thank you for your feedback. We have updated the manuscript to specify the resolution of the CNES-CLS22 MDT as 7.5′. Additionally, we have included the order of the XGM2019e model for completeness. These additions enhance the clarity and accuracy of the model descriptions.

Changes in manuscript: Page 4.

9. P5L129: The term "reference data" is inappropriate in this context and should be revised to a more suitable term.

**Response:** We have revised the text to clarify the specific objectives of using the GEBCO model: first, to assess the performance of SDUST2023VGGA in different seafloor topographies, and second, to examine the correlation between SDUST2023VGGA and the GEBCO model, particularly in terms of vertical gravity gradient anomalies and their ability to explain seafloor topography. This revision directly addresses the study's aims without the use of the term "reference data."

Changes in manuscript: Line 129, Page 5

10. P5L139-155: Several uncommon terms are used in this section. Consider replacing "several" with "multiple" and "set to" with "defined as" to adopt a more formal tone. Additionally, substituting "Clip" with "Crop" would improve clarity and understanding.

    **Response:** Thank you for these helpful suggestions. We have implemented the following changes to improve clarity and formality:

    a. We have replaced "several" with "multiple" to adopt a more formal tone.

    b. We have replaced "set to" with "defined as" for better precision and consistency.

    However, we have retained the term "Clip" instead of substituting it with "Crop," as "Clip" is widely recognized in geospatial analysis for describing the exclusion of data beyond a defined boundary. Using "Crop" could introduce ambiguity by implying a visual adjustment rather than a computational operation. We appreciate your understanding of this decision and your valuable input.

    We appreciate your understanding and thank you again for your valuable feedback.

    Changes in manuscript: Lines 139-155, Page 5.

11. P6L155: The use of "initially" creates confusion and hinders readability. This phrasing should be revised for greater clarity.

    **Response:** Thank you for pointing this out. We have removed the term "initially" to eliminate potential confusion, resulting in a clearer and more straightforward explanation of the model's construction.

    Changes in manuscript: Line 155, Page 6.

12. P7L164: The variable "h" in Equation 1 is not defined. Please provide a clear definition or description of this variable.

    **Response:** We thank the reviewer for identifying the omission of the definition for the variable h in Equation (1). Clearly defining all variables is essential for the manuscript's clarity and comprehensibility.

    Changes in manuscript: Line 164, Page 7.

13. P7L170: Since $\nabla^2 U = 0$, the vertical component can be expressed using the two horizontal components. Therefore, the original phrasing is unsatisfactory and needs to be improved to accurately convey this relationship.

    **Response:** Thank you for your observation. We have revised the phrasing to clearly explain how the Laplace equation constrains the vertical gravity gradient anomaly in terms of the geoid and its horizontal derivatives. The revised text explicitly describes the relationship between the vertical and horizontal components, ensuring accurate communication of this concept.

    Changes in manuscript: Line 170, Page 7.

14. P7L175: The rationale behind increasing the resolution should be explained to justify this methodological choice.

**Response:** We appreciate the reviewer's comment. We have added an explanation in the manuscript to justify the increase in resolution, emphasizing its importance in enhancing the spatial detail and accuracy of the modeled vertical gravity gradient anomaly. This addition clarifies the reasoning behind our methodological choice.

Changes in manuscript: Line 175, Page 7.

15. P7L181: Replace "original" with "full wavelength" to enhance the reader's understanding of the "remove-restore" process.

**Response:** Thank you for the suggestion. We have replaced "original" with "full-wavelength" in the manuscript to clarify the remove-restore process and improve the overall accuracy of the description.

Changes in manuscript: Line 181, Page 7.

16. P8L193: Please verify the accuracy of this citation, as the manuscript does not appear to include any analysis regarding the calculation of window size and its impact on the computation.

**Response:** We appreciate the reviewer pointing this out. After reviewing the manuscript, we confirmed that it does not include an analysis of window size calculation or its computational impact. As such, we have removed the citation to maintain accuracy and relevance.

Changes in manuscript: Line 193, Page 8.

17. P9L203: We recommend referring to it as "component" rather than "direction" to maintain consistency and precision in terminology.

**Response:** Thank you for the recommendation. We have revised the manuscript to replace "direction" with "component" to ensure consistency and precision, aligning with standard geophysical terminology.

Changes in manuscript: Line 203, Page 9.

18. P9L205: Based on your description, the process at this point should involve calculating the residual geoid rather than the geoid. Please verify and correct this accordingly.

**Response:** You are correct that at this stage in the process, we should be calculating the residual geoid rather than the geoid itself. We have reviewed the manuscript and corrected the relevant section to reflect the calculation of the residual geoid. We have updated the manuscript accordingly to ensure accuracy and consistency with the described process.

Changes in manuscript: Line 205, Page 9.

19. P9L207: The variable "l" still needs to be defined. Additionally, the restore process should be

clarified. The explanation of the remove-restore process in other sections is more detailed than in the methods section, which is inconsistent and needs to be addressed.

**Response:** Thank you for highlighting these points. We have made the following revisions:

a. Defined l as the horizontal distance between two points in the calculation.

b. Clarified the restore process to align with the detailed explanation provided in other sections, ensuring consistency and precision in describing the remove-restore methodology.

Changes in manuscript: Line 207, Page 9; Methodology.

20. P9L218: The original text refers to the second-order partial derivative of the MSS data. It should clarify whether this pertains to the second-order partial derivative of the geoid or the residual geoid.

    **Response:** Thank you for identifying this ambiguity. The calculation pertains to the geoid height, not the MSS or residual geoid directly. We have revised the text to clarify this, ensuring it accurately reflects the process of computing geoid height and its second-order partial derivatives.

    Changes in manuscript: Line 218, Page 9.

21. P10 Fig 4: Does "mean" in the flowchart refer to the operation on the overlap section? Additionally, please provide a clear definition of "overlap" within the manuscript to ensure clarity.

    **Response:** We have added a detailed explanation of the trimming and merging processes for overlapping partitions in the manuscript. Figure 4 has also been updated to clarify the use of "mean" in relation to the overlap section.

    Changes in manuscript: Page 10; Figure 4; Methodology.

22. P14: The author applied two rounds of filtering to the residuals for data analysis. It should be clarified whether the filtered data were used in subsequent comparisons and processing to understand the impact of this filtering on the results.

    **Response:** Thank you for pointing this out. We have updated the manuscript to explicitly clarify the purpose of the filtered data and its role in subsequent comparisons and analyses. This differentiation ensures the impact of the filtering process on the results is well understood.

    Changes in manuscript: Page 14; Results section.

23. P15, "Discussion". The SIO has released SWOT refined VGG data, the authors can compare your own data with the SIO's product. I suggest the authors to add these results regarding the comparisons with SWOT-derived VGG.

    **Response:** Thank you for this insightful suggestion. In response, we have added a comparison between our results and the SWOT-derived vertical gravity gradient (VGG) data from SIO in

the "Discussion" section. This comparison validates our data and highlights key similarities and differences. Additionally, we have extended the discussion by incorporating an analysis of the relationship between our results and GEBCO bathymetry.

Changes in manuscript: Page 15; Discussion section.

24. P15L324-356: The discussion is overly verbose, which impacts readability. The text should be reorganized for clarity and conciseness, focusing on the most relevant points and eliminating redundant information.

**Response:** Thank you for the valuable feedback. We have reorganized the discussion section to enhance clarity and readability. Redundant information has been removed, and the revised text focuses on the most relevant points, ensuring a concise and precise presentation of the results and their implications.

Changes in manuscript: Lines 324-356, Page 15; Discussion section.

25. P19L375-379: The results appear to be correct; however, there is a concern that when the slope is below 1%, the STD and RMS values appear to be somewhat larger. Please provide possible explanations for this observation.

**Response:** Thank you for your insightful comment. In low-slope areas, the residual geoid model is more sensitive to noise and systematic errors in the input data. This heightened sensitivity may lead to slightly larger STD and RMS values, even though these areas are typically expected to exhibit higher model stability. We have revised the manuscript to include this explanation.

Changes in manuscript: Lines 375-379, Page 19; Results section.

26. P20: For Region C in Fig. 9, was land data included in the calculation? The color bar suggests that this region has complex terrain and areas extending above sea level, which could affect the calculation results. If land data was used, the analysis in this section should be revisited to account for these factors.

**Response:** Thank you for the observation. Land data was not included in the calculation. All experimental data were processed using GMT to remove land areas before the analysis. This ensures the results are based solely on oceanic regions. Additionally, the analysis across different regions confirms consistency and excludes any interference from land areas.

27. P22L444-L449: The relationship between VGGA and seabed topography is not linear. Consider removing the Linear Regression analysis unless it is included for comparison with other methods. Please evaluate the primary purpose of this analysis when making your decision.

**Response:** Thank you for your feedback. After reconsidering the role of linear regression, we acknowledge its limitations in capturing the nonlinear relationship between the GEBCO data

and the VGGA model. However, we have retained it as a baseline comparison to highlight the inadequacies of simpler models. The revised manuscript emphasizes the superior performance of non-linear methods, such as SVR and MLP, which better capture the complex relationships between the VGGA model and bathymetric data.

Changes in manuscript: Lines 444-449, Page 22; Discussion section.

28. P25: The conclusion section lacks depth, and some analyses, such as those of SVR and MLP, are not well summarized. Expanding this section to include a comprehensive summary of all key findings would strengthen the manuscript.

    **Response:** We have revised the conclusion section to provide a more comprehensive summary of key findings, including specific results from the SVR and MLP analyses. This revision eliminates redundancy and strengthens the synthesis of the study's findings, offering a robust and concise conclusion.

    Changes in manuscript: Page 25; Conclusion section.

29. P27: The format of the References needs to be standardized. Please ensure that all references adhere to the journal's formatting guidelines for consistency and professionalism.

    **Response:** The references have been revised to adhere to the journal's formatting guidelines, ensuring consistency and professionalism throughout. Additionally, some less relevant references have been removed to maintain the focus and relevance of the cited literature.

    Changes in manuscript: Page 27; References section.

After addressing all the reviewers' comments, we believe that the revised manuscript has been significantly improved and now meets the high standards of your journal. We sincerely appreciate the time and effort you have dedicated to reviewing our work.

Sincerely,

Jinyun Guo